# Synoptic scale variability of surface winds and ocean response to atmospheric forcing in the eastern Austral Pacific Ocean

Iván Pérez-Santos[1, 2], RomanetSeguel[2, 3, 4], Wolfgang Schneider[5, 6], Pamela Linford[7], David Donoso[5,8], Eduardo Navarro[3, 6], Constanza Amaya-Cárcamo[9], Elías Pinilla[4], Giovanni Daneri[10].

[1]Centro i-mar de la Universidad de los Lagos, Puerto Montt, Chile
[2]Centro de Investigación Oceanográfica COPAS Sur-Austral, Universidad de Concepción, Chile
[3]Programa de Postgrado en Oceanografía, Departamento de Oceanografía, Universidad de Concepción, Chile
[4]Instituto de Fomento Pesquero, Chile
[5]Departamento de Oceanografía, Campus Concepción, Universidad de Concepción, Chile
[6]Millennium Institute of Oceanography (IMO), University of Concepción, Chile
[7]Programa de Doctorado en Ciencias mención Conservación y Manejo de Recursos Naturales, Centro i-mar, Universidad de Los Lagos, Puerto Montt, Chile.
[8]Pontificia Universidad Católica de Valparaíso, Chile.
[9]Departamento de Geofísica, Universidad de Concepción, Chile.
[10]Centro de Investigaciones en Ecosistemas de la Patagonia, Coyhaique, Chile.

*Corresponding author: Iván Pérez-Santos: ivan.perez@ulagos.cl*

**Abstract**. In the southern hemisphere, macroscale atmospheric systems such as westerly winds and the Southeast Pacific subtropical anti-cyclone (SPSA) influence the wind regime of the eastern Austral Pacific Ocean. The average and seasonal behaviors of these systems are well-known, although wind variability at different time and distance scales remains largely unexamined. Therefore, the main goal of this study was to determine the variabilities of surface winds on a spatiotemporal scale from 40°–56° S, using QuikSCAT, ASCAT, and ERA5 reanalysis surface-wind information complemented with in situ meteorological data. In addition, interactions between the atmospheric systems, together with the ocean-atmosphere response, were evaluated for the period 1999–2018. The empirical orthogonal function detected dominance at the synoptic scale in mode 1, representing approximately 30 % of the total variance. In this mode, low and high atmospheric pressure systems characterized wind variability for a16.5-day cycle. Initially, mode 2—which represents approximately 22 % of the variance—was represented by winds from the west/east (43°–56° S), occurring mostly during spring and summer/fall and winter at an annual time scale (1999–2008) until they were replaced by systems cycling at 27.5 days (2008–2015).This reflects the influence of the baroclinic annual mode in the Southern Hemisphere. Mode 3, representing approximately 15 % of the variance, involved the passage of small-scale low and high atmospheric pressure (LAP and HAP) systems throughout Patagonia. Persistent Ekman suction occurred throughout the year south of the Gulf of Penas and beyond the Pacific mouth of the Magellan Strait. Easterly Ekman transport (ET) piled these upwelled waters onto the western shore of South America when winds blew southward. These physical mechanisms were essential in bringing nutrients to the surface and then transporting planktonic organisms from the oceanic zone to Patagonian fjords and channels. In the zonal band between 41°–43° S, the latitude of Chiloe Island, upward Ekman pumping and Ekman transport during

spring and summer favored a reduced sea surface temperature and increased chlorophyll-a levels; this is the first

time that such Ekman upwelling conditions have been reported so far south in the eastern Pacific Ocean. The influence of the northward-migrating LAP systems on the ocean-atmosphere interphase allowed us to understand, for the first time, their direct relationship with recorded nighttime air temperature maxima (locally referred to as "nighttime heatwave events"). In the context of global climate change, greater attention should be paid to these processes based on their possible impact on the rate of glacier melting and on the Austral climate.

**Keywords:** Atmospheric pressure systems, Ekman upwelling, Pacific Ocean, Patagonia, synoptic scale

## 1.    Introduction

The eastern Austral Pacific Ocean (40° to 56° S) is under the influence of westerly winds and the Southeast Pacific subtropical anti-cyclone (SPSA) (Tomczak and Godfrey, 1994; Stewart, 2002). In this region, westerly winds are stronger than those in the northern hemisphere on average and extend in a belt from 40°–60° S (Talley et al., 2011).

The SPSA shows an annual migration, reaching its southern position (~40°–46° S) in the austral summer, owing to the poleward displacement of the intertropical convergence zone (Rahn and Garreaud, 2013; Ancapichun and Garcés-Vargas, 2015; Schneider et al., 2017). The anti-clockwise rotation of winds from the SPSA generates northerly winds along the coastlines of Chile and Peru, contributing to the maintenance of upwelling conditions all year and producing one of the highest productivity marine ecosystems in the world (Kampf and Chapman, 2016).

The system is well known for the contribution of the westerly winds and the SPSA to the circulation regimen (e.g., the formation of the Humboldt Current system) (Thiel et al., 2007; Fuenzalida et al., 2008).

While most studies have focused on the behavior of the SPSA (Rahn and Garreaud, 2013; Ancapichun and Garcés-Vargas, 2015; Schneider et al., 2017) and the effect of oceanic-atmospheric interaction, little is known about the variability in the surface winds in the eastern Austral Pacific Ocean. Therefore, the principal goal of this study was

to determine the spatiotemporal variability of the surface winds that extend from 40° S–56° S using different satellite wind products, reanalysis climate date sets, and in situ meteorological information. The interaction between the Austral Pacific surface-wind regimen and the SPSA was also considered with oceanic-atmospheric dynamics.

The principal hypotheses of our study were: (1) the passage of synoptic-scale atmospheric events throughout the eastern Austral Pacific Ocean, such as low and high atmospheric pressure (LAP and HAP) systems, dominate the

surface-wind variations in the study area; and (2) the interaction between synoptic-scale atmospheric events, such as the SPSA with LAP systems, allows the advection of warm air over Patagonia and creates maximum surface air temperatures at nighttime, especially in fall and winter.

In terms of time-distance scales, atmospheric systems have been categorized as macro-,meso-, and microscale (Orlanski, 1975; Ray, 1986; Holton, 1992). The macroscale definition is divided into planetary and synoptic scales.

Winds that impact the globe belong to the planetary scale, such as the westerly and trade winds and El Niño (Tomasz, 2014), extending over distances from 1000–40000 km, with time scales of weeks or longer. Synoptic scale systems cover approximately 100–5000 km, with a timescale of days to weeks and include events such as atmospheric pressure systems like subtropical anti-cyclones and hurricanes. Mesoscale events cover a distance of

1−100 km and timescale of minutes to hours and include events such as thunderstorms, tornadoes, and sea/land

breezes, while microscale systems cover a range of <1 km and include events such as turbulence, dust devils, and gusts, occurring in seconds to minutes. In this manuscript, we evaluate the oceanic response to synoptic scale atmospheric events, including LAP and HAP systems. From a meteorological point of view, in a LAP system winds rotate clockwise (southern hemisphere) around a core of low pressure and are generally associated with severe weather conditions (e.g., intense wind, rain, and clouds). In contrast, in a HAP system, winds rotate

counterclockwise (southern hemisphere), and high pressure is located in the center of the event, producing mostly good weather conditions with clear skies.

The passage of LAP events throughout Patagonian fjords and channels, such as Puyuhuapi Fjord, creates intense vertical mixing that favors microalgal blooms, increasing primary production during the winter season to reach a magnitude similar to the traditionally productive spring-summer season (Montero et al., 2017). In the case of HAP

events, which produce alongshore winds (northward), the contribution to the upwelling conditions (offshore Ekman transport) in the northern part of the eastern Austral Pacific coast line has not been quantified. Similarly, LAP systems also produce alongshore winds, although in the opposite direction (southward), and favor downwelling; the mechanisms and effects of these events are addressed in Sect. 3.2.

In the California upwelling systems (32°-44° N), the contribution of Ekman transport (ET) and Ekman pumping

(EP), with upward velocities favoring upwelling and primary production and downward velocities contributing to downwelling, were quantified using an atmospheric model, finding that EP was more important to the processes than ET was (Pickett and Paduan, 2003). In the central-northern region of Chile, where only ET has previously been evaluated as the leading contributor to upwelling (Sobarzo and Djurfeldt, 2004), one study (Bravo et al., 2016) has demonstrated that EP contributed to 40–60 % of the total upwelling transport. ET and EP, derived here using

surface-wind products from the QuikSCAT and ASCAT satellites and the ERA5 reanalysis climate data set, were used to quantify the contributions of these processes to the total upwelling, with special attention to the offshore region of Chiloe Island (42°–43.5° S), where northward wind occurs in spring-summer due to the SPSA influence.

In this manuscript, statistical analyses including empirical orthogonal function (EOF) using the surface-wind products from the QuikSCAT and ASCAT satellites and the ERA5 reanalysis from 1999−2015 allowed the

estimation of the importance of synoptic-scale events in the wind variability of the eastern Austral Pacific Ocean. In addition, the ocean-atmosphere response to the surface wind was evaluated using reanalysis data to the present day, in combination with a time series for chlorophyll-a, fluorescence, and sea surface temperature data from the MODIS-Aqua satellite. Air temperature and atmospheric pressure from ERA5 and in-situ data from buoy and meteorological stations were also included in the analyses.

**2.    Materials and methods**

**2.1    Satellite and reanalysis surface-wind data**

Surface-wind data were obtained from SeaWinds scatterometers mounted on the QuikSCAT and ASCAT satellites. QuikSCAT wind vectors were obtained daily in a 0.5° × 0.5° grid (http://www.ifremer.fr). The root mean square

errors (RMSE) for wind velocity and direction were specified to be less than 1.9 m s$^{-1}$ and 17°, respectively (Piolle and Bentamy, 2002). Analysis of QuikSCAT satellite wind data covered the period from July 1999 to November 2009. For the ASCAT wind product, the temporal resolution was also daily-averaged at a spatial resolution of 0.25° × 0.25° over two swaths with widths of 550 km each (Bentamy et al., 2008). The ASCAT data were validated with moored buoys from the National Data Buoy Center (NDBC), MF-UK (Météo-France and UK Met office), TAO buoys, QuikSCAT scatterometers, and the European Centre for Medium-Range Weather Forecasts (ECMWF). Comparison between ASCAT and QuikSCAT data demonstrated good agreement for a wind speed range of 3–20 m s$^{-1}$; however, ASCAT underestimated wind speeds outside this range. The RMSE was not uniform worldwide, with values of 1.5–3.5 m s$^{-1}$at high latitudes compared to a global average of  2 m s$^{-1}$ from a wind direction of 18° (Bentamy et al., 2008). Furthermore, comparison of ASCAT wind fields with data from moored buoys had correlation coefficients of 0.86 with RMSE values of  2 m s$^{-1}$ (Bentamy and Croize-Fillon, 2011). In this study, the ASCAT product was used for the period April 2007 to December 2015.

The ERA5 reanalysis climate data set of surface wind was added to the manuscript because it offered continued surface-wind data with high temporal and spatial resolution from 1979 to the present day (https://cds.climate.copernicus.eu). Data covering the period from 1999 to 2018 was included. The ERA5 reanalysis used 4D-Var data assimilation in CY41R2 from the ECMWF with 137 levels in the vertical and the top level at 0.01 hPa. This data set is available in hourly temporal resolution with a regular spatial grid of 0.25° × 0.25°. Hourly surface-wind data were averaged daily for the analyses presented in Figures 2–10. As surface-wind input, ERA5 incorporated different satellite scatterometer data such as AMI (ERS 1 and ERS2), ASCAT (METOP-A/B), OSCAT (OCEANSAT-2), and SeaWinds (QuikSCAT). Furthermore, in situ data provided by the World Meteorological Organization information system (WMO-WIS) (e.g., land stations, drifting buoys, ship stations, radiosondes, radars, and aircraft data) was also added. To understand the origin and influence of the "nighttime heatwave events" in Patagonia, the air temperature (2 m) and surface atmospheric pressure from ERA5, with hourly temporal resolution, were utilized.

Local and regional validation process analyses were performed using data from Navy lighthouses (NLHs) located in the coastal zone (Fig. 1). The data from the NLHs covered the periods2009–2011 and 2014, when the operation of the QuikSCAT, the ASCAT, and the ERA5 surface-wind products coincided. Additionally, data from an oceanographic buoy moored in Reloncaví Sound (Fig. 1) was used for comparison with the ERA5 reanalysis from 2017–2018.A Taylor diagram was applied to all data sets as a validation tool (Taylor, 2001). In general, the validation between satellites and reanalysis surface-wind products with the in-situ wind data demonstrated satisfactory results, with correlation coefficients of 0.5–0.9 and RMSE and standard deviations of ~2–4 m s$^{-1}$. The ERA5 data showed the highest statistical results among the surface-wind products (see supplementary material, Fig. S1–S5, for further details regarding the validation process).

## 2.2    Environmental data from buoys and meteorological stations

Data from an oceanographic buoy installed in the northern section of Puyuhuapi Fjord (Fig. 1, 44°35.3' S, 72°43.6' W),equipped with atmospheric (wind speed and direction, air temperature, and atmospheric pressure), and surface

water (temperature and conductivity) sensors, was used to understand the fjord-atmosphere interactions. The raw
       atmospheric data from the buoy were collected with a temporal resolution of 3 min, and the water data were
       registered hourly at depth of ~1 m. The time series from the oceanographic buoy started in April 2011 and finished
       in July 2013. A meteorological station was installed on the coast, ~500 m from the buoy, to continue atmospheric
       measurements in this region. The meteorological station registered raw atmospheric data every 15 min (wind speed
and direction, air temperature, and atmospheric pressure) from April 2014 to August 2017. Generally, all the
       atmospheric data from the buoy was temporally homogenized, and the data from the meteorological stations were
       averaged hourly.

### 2.3    Satellite-derived sea surface temperature, chlorophyll-a, and fluorescence data

       Satellite-derived images and a time series of chlorophyll-a (Chl-a) concentrations, normalized fluorescence line
heights (FLH), and sea surface temperature (SST) were obtained from the moderate resolution imaging
       spectroradiometer (MODIS) sensor on the Aqua satellite. The data were obtained with a spatial resolution of 4 $km^2$
       per pixel, at nadir, over cloud-free ocean areas, with a temporal resolution of 8 days, covering the period from
       2002–2018. Chl-a, FLH, and SST images and time series were extracted from the geospatial interactive online
       visualization and analysis infrastructure (Giovanni; https://giovanni.gsfc.nasa.gov) and used as measures of the
marine response to the surface winds and the associated processes(e.g., ET and EP).

### 2.4    Derived variables

       The influence of surface winds on the ocean response was monitored throughout the calculation of the ET and EP.
       Both processes participate in the injection of nutrients from the deep layer to the euphotic zone, where the
       phytoplankton are more abundant, increasing the primary biological production (Thurman and Trujillo, 2004). In the
ET, the wind blowing toward the equator (Polar) on the western coastline generates an offshore (onshore) mass
       transport to the ocean, causing the upwelling (downwelling) of rich water. In contrast, EP originates from the
       divergence (convergence) of wind stress curl, contributing to the upwelling (downwelling) of water due to the
       positive (upward) and negative (downward) EP velocities (Tomczak and Godfrey, 1994; Stewart, 2002).

       Using QuikSCAT and ASCAT surface-wind data, the components of the zonal and meridional wind stress ($\tau_u$ and
$\tau_v$, respectively) were calculated as shown in Eq. (1):

$$\tau_u = \rho_a C_d u U_{10}, \ \tau_v = \rho_a C_d v U_{10} \qquad (1)$$

       In Eq. (1), $\rho_a$ is air density (1.2 kg $m^{-3}$), $C_d$ is a dimensionless drag coefficient; $u$ and $v$ are the zonal and meridional
       wind components, respectively; and $U_{10}$ is the magnitude of the wind vector 10 m above sea level. $C_d$ was calculated
       using the formula proposed by Yelland and Taylor (1996), in which the coefficient varies as a function of the wind
velocity, according to Eqs. (2 and 3):

$$C_d = 0.29 + \frac{3.1}{U_{10}} + \frac{7.7}{U^2_{10}} \times 10^{-3}, \text{ for } U_{10} \leq 6ms^{-1} \qquad (2)$$

$$C_d = 0.60 + 0.070U_{10} \times 10^{-3}, \text{ for } 6ms^{-1} \leq U_{10} \leq 26ms^{-1} \qquad (3)$$

Ekman surface transport, M (m$^2$ s$^{-1}$), was calculated for each grid point of the satellite wind field using Eq. (4) below (Smith, 1968):

$$\vec{M} = \frac{\vec{\tau}}{\rho_w f} \qquad (4)$$

where $\vec{\tau}$ is the wind stress vector, $\rho_w$ is the water density (1025 kg m$^{-3}$) and $f$ is the Coriolis parameter. The EP velocity, W$_E$ (m$^3$ s$^{-1}$), was calculated according to Eq. (5) (Smith, 1968):

$$W_E = \frac{1}{\rho_w f} \nabla \times \vec{\tau} \qquad (5)$$

where $\nabla \times \vec{\tau}$ is the wind stress curl, which was derived by first-order cross-differencing of the wind stress field.
This implies that no curl computation was possible for the grid points nearest the coast. This drawback was overcome by applying coKriging to the wind stress curl in two dimensions, which in turn allowed extrapolation toward the coast (Marcotte, 1991).

To quantify the relative importance of EP for the total upwelling transport (TUT), EP velocities were integrated up to ~50 km offshore along three transects located in the northern (42.7° S), central (47.2° S), and southern (52.0° S)
parts of the study region (Fig. 1). This calculation was performed to obtain the vertical transport (m$^3$ s$^{-1}$) for each selected transect and compare it with the ET obtained using Eq. (4), following the methodology proposed by Pickett and Paduan (2003). In Fig. 11 TUT was averaged every 8 days for comparison with the MODIS-Aqua variables (e.g., Chl-a, FLH, and SST).

### 2.5    Data analysis

Zonal and meridional surface winds from QuikSCAT (1999–2009), ASCAT (2007–2015), and ERA5 reanalysis (1999–2015) were used to apply EOF analysis (Emery and Thomson, 1998; Kaihatu et al., 1998) to determine the modes of variability that dominated the spatiotemporal behavior of the wind field in the eastern Austral Pacific Ocean. Before computing the EOFs, long-term means and linear trends were removed for each scatterometer (QuikSCAT and ASCAT) and the reanalysis product (ERA5) separately. To complete this process, the mean and
linear trend calculations were applied to all grid points covering the entire data set period.

A Morlet wavelet analysis was applied (Torrence and Compo, 1998) to the time-dependent coefficients of the three leading modes, resulting from real-vector EOF analysis of the QuikSCAT, ASCAT, and ERA5 reanalysis surface-wind fields. This wavelet analysis allowed for the distinction of time and duration of the dominant periods of the different atmospheric processes. The wavelet spectra were used to calculate the time-averaged spectra for the entire
sampling period and are subsequently referred to as the global wavelet spectrum (Torrence and Compo, 1998).

### 3. Results

### 3.1 Surface-wind features and variability

Analysis of the surface-wind, long-term daily mean, for the period 1999–2015, using the QuikSCAT and ASCAT satellite products and the ERA5 reanalysis climate data set, showed similar patterns (Fig. 2). In general, westerlies
were the predominant surface winds, especially between 42° and 45° S, although a more detailed analysis indicated different features. First, north of 42° S, the wind was slightly west-southwesterly. Second, south to 45° S, the wind started an inclination from the west to the northwest direction, and third, between 52° S and 56° S, the wind blew along the Austral coast of the Magellan region, while in the rest of the study area, the wind direction was perpendicular to the coast. The surface-wind average registered as a meridional gradient, in which low speeds (5−6
m s$^{-1}$) were observed in the northern domain, and stronger winds (10–2 m s$^{-1}$) were registered closer to 51° S. The standard deviations were similar between the satellite products (±3.0 to ±4.2 m s$^{-1}$), representing the same meridional gradient observed in the surface-wind magnitude, but the ASCAT data registered a lower variability and less intense surface-wind magnitudes, compared with the data obtained by QuikSCAT (Figs. 2a and 2b). Nevertheless, lower standard deviations and wind magnitudes were obtained by the ERA5 reanalysis data (Fig. 2c).
Computations of the seasonal cycle, using all datasets (e.g., QuikSCAT, ASCAT and ERA5), showed a similar meridional gradient to that obtained in the average analysis, highlighting the time-persistence and high intensity of the northwesterly winds in the open ocean water of the Magellan region (51° to 56° S).

EOF analysis allowed further understanding of the surface-wind variability modes and distribution of the total variance. The EOF for the QuikSCAT (1999−2009) daily data showed a concentration of ~70 % of the total variance
in the first three empirical modes: EOF-1=30.01 %; EOF-2=22.5 %; EOF-3=16.4 % (Fig. 3a−c). For the equivalent ASCAT (2007−2015) daily wind data, the EOF represented ~65 % of the total variance, in the first three empirical modes: EOF-1=27.9 %; EOF-2=22.5 %; EOF-3=15.3 % (Fig. 3d−f). In contrast, the ERA5 reanalysis data showed similar variances but covered the total sampling period with EOF-1=28.6 %, EOF-2=25.9 % and EOF-3=18.2 % (Fig. 3g−i). The spatial structure of the first three modes from the QuikSCAT, ASCAT, and ERA5 databases were
similar (Fig. 3). In terms of the spatial structure of mode 1 (Fig. 3a, d and g), southerly and southwesterly winds dominated the study area when the time-dependent coefficient was positive (Figs. 4a, 5a and 6a, PC-1). When principal component 1 (PC-1) was negative, the spatial structure of mode 1 rotated, and northerly and northeasterly winds occurred.

The global spectrum analysis performed for PC-1 denoted the dominant 16.5-day cycle (Figs. 4b and 5b). The PC-1
monthly mean calculation demonstrated that southerly winds occurred mostly during the fall and spring, while northerly winds were more frequent during the winter, spring, and summer (Figs. 4c and 5c). The global spectrum of the complete and continuous ERA5 reanalysis data set again showed the 16.5-day cycle and the approximately 314-day annual cycle (Fig. 6b). In the monthly mean calculations, observations showed an alternating dominance of southerly and northerly winds (Fig. 6c). The southerly and northerly winds were associated with the passage of
intense HAP (Fig. 7a) and LAP (Fig. 7b) systems throughout the study region.

The spatial structure of mode 2 highlights the presence of the easterly (positive time-dependent coefficient) and westerly winds (negative time-dependent coefficient) (Fig. 3b, 3e, and 3h). The global spectrum for the PC-2 (Fig. 4d and 4e) represented the dominance of the 374-dayannual cycle for the QuikSCAT database. The low-pass filtered time series for the PC-2 (Fig. 4d, red line) showed the occurrence of the most positive values during fall and winter,

represented by the easterly winds (Fig. 4f). The negative part of the PC-2 was observed during spring and summer, highlighting the presence of the westerly winds (Fig. 4f). Even though the spatial structure of mode 2 from the ASCAT database presented a similar pattern to QuikSCAT mode 2, the annual cycle period was not detected in the spectrum of PC-2 (Fig. 5d and 5e). In this analysis, 27.5-day and 16.5-daycycles were obtained. The monthly mean for PC-2 coincided with the results from QuikSCAT during winter (easterly winds) and spring (westerly winds), but

was different in summer and fall, when the wind direction varied (Fig. 5f). For PC-2, the global spectrum signal and monthly mean calculations, obtained using the ERA5 data, reflected a combination of results registered with the QuikSCAT and ASCAT data sets, highlighting the dominance of the 374- and 27.8-day cycles (Fig. 6d–f). Figures 7c and 7d show examples of the atmospheric systems involved in the wind direction variability characteristic of this mode.

The spatial structure of mode 3 can be represented by a clockwise atmospheric circulation (e.g., Fig. 7e) of surface winds in the same direction as LAP system when the time-dependent coefficient was positive (Figs. 3c, 3f, and 3i). The rotation of the winds became counterclockwise (e.g., Fig. 7f) when the time-dependent coefficient was negative, representing a structure similar to that seen for a HAP system (Figs. 4g, 5g, and 6g). Periods of 2−8 days were detected in the spectrum analysis of all data sets (QuikSCAT, ASCAT, and ERA5), while semiannual (157 days)

and annual cycles were also observed (Figs. 4h−i, 5h−i and 6h−i).

To capture the influence of the LAP and HAP systems in the EOF patterns, the ERA5 data set was used to carry out a further EOF analysis, but this time the study region was expanded to the west (120° W) and the north (30° N). This EOF analysis confirmed our previous conclusions (See Supplementary Material, Fig. S6–S7).

Wavelet analysis facilitated the observation of the year-round dominance of the synoptic time scale obtained by PC-

1 (Fig. 8a and 8b). An evident change in time scales was observed for PC-2, (e.g., the annual cycle dominated from 2000−2008) (Fig. 8c), but from 2009−2015, a 20−30-day cycle was more intense than the annual cycle (Fig. 8d). In PC-3, the semiannual signal observed in the global spectrum (Fig. 4h) occurred in 2004 (Fig. 8e). However, while the annual cycle registered in Fig. 5h was clear in 2011 (Fig. 8f), synoptic time scales were persistent from 2000– 2015 (Fig. 8e and 8f). The wavelet analysis performed on the ERA5 PC-1, PC-2, and PC-3 (Fig. 8g−i) confirmed the

results obtained from the QuikSCAT and ASCAT data sets, showing the change in time scales registered in PC-2 starting in 2009 and higher energies from the annual cycles from 1999 to 2006.

## 3.2    Derived parameters from surface winds and ocean implications

The average dominance of the westerly surface-wind stress generally produced a northerly Ekman transport (ET) in the study region (Fig. 9a−c). On average, the ET ran parallel to the coast, from 40°−47° S and to 56°S.The

inclination of the coastline and the influence of westerly wind stress contributed to the change in the ET direction,

orienting mostly perpendicular to the coast. This region (48°–56° S) demonstrated the highest ET value (2.16 m$^2$ s$^{-1}$) recorded due to the presence of the most intense regional winds (Fig. 2). Moreover, a wide area of positive (upward motion) and maximum Ekman pumping (EP=0.25 m day$^{-1}$) was observed at approximately 51° S in the QuikSCAT data; the positive EP extended across the study area (Fig. 9a).The same area of positive and intense EP was observed in the ASCAT database; however, in the northern part of the study region, between 40° S–48° S, the upward EP was located closer to the coast and covered approximately the first 100 km (Fig. 9b). The long-term mean of daily ET and EP, calculated with ERA5, was similar to that obtained for the QuikSCAT period, showing the greatest concurrences with areas where EP was maximized (Fig. 9c). However, the ERA5 values were higher when EP=0.57 m day$^{-1}$ at 50.5° S–76.25° W. The EP was also high in the ERA5 dataset along the coastline between 40° S–44° S (Fig. 9).

The analysis using QuikSCAT and ASCAT (from Figs.2−9) and the reanalysis product demonstrated that ERA5 showed stronger similarities in the results. Hence, to understand the annual variability of the ET and EP and the contribution of both processes to the total upwelling transport (TUT), three data time series from ERA5 were extracted from 1999–2018 for the northern, central, and southern parts of the study region (Fig. 10). In the northern part of the study region, the daily mean long-term TUT in the ocean off the coast of Chiloe Island showed high variability (±0.82 m$^3$ s$^{-1}$) year-round, especially during the fall and winter, when onshore ET dominated the TUT. This changed during part of the spring and throughout the summer, when mainly offshore ET dominated, but the magnitude was weaker than that observed in winter. The EP was positive and dominated the TUT (Fig. 10a). The long-term monthly mean of the time series showed the dominance of downwelling conditions from May to October (Austral fall-winter). The upwelling typically began in November and finished in April, with a significant contribution from the EP (Fig. 10b). The cumulative transport was generally favorable to downwelling from May to December, with reduced upwelling in the summer (Fig. 10c).

In the time series data for the Gulf of Penas, a year-round variability was observed in the long-term daily mean of the TUT (±0.97 m$^3$ s$^{-1}$), but offshore ET events decreased and EP showed reduced positive values (Fig. 10d). Downwelling conditions prevailed due to the dominance of the ET during the year (Fig. 10e) and the cumulative transport was negative (downwelling) for ET and TUT and higher than observed in the northern time series (Fig. 10f).

In the southern part of the study region, close to the entrance of the Magellan Strait, the absolute maximum (-8.25 m$^3$ s$^{-1}$) was reported, along with higher variability of the TUT (±1.24 m$^3$ s$^{-1}$), which was dominated by the ET. The EP was positive and favorable for upwelling but less intense than for the ET (Fig. 10h). The long-term monthly mean for transport showed the highest values for the ET and the highest contribution of this process to the TUT, even though the EP was positive and favorable for upwelling (Fig. 10i). The cumulative transport was also the most important compared with the other time series (Fig. 10j).

Downwelling conditions generally dominated the study region, but in the open ocean water around Chiloe Island, upwelling was observed during spring-summer owing to the contribution of the wind stress curl that generated positive EP velocities. Considering the previous results showing that surface winds (ET and EP) contributed to the injection of subsurface water to the surface layer, the time series of TUT, together with satellite data for Chl-a, FLH,

and SST, were used to evaluate the oceanic response to favorable upwelling conditions (Fig. 11).The time series of TUT from 2002 to 2018 showed an annual cycle in which upwelling conditions occurred during spring and summer(Fig. 11a, red shaded area), while downwelling conditions were typically observed in the fall and winter (Fig. 11a, blue shaded area). The Chl-a anomalies showed a positive response to the TUT (Fig. 11b) with a correlation coefficient (Corr. coef.) of 0.32. Because the Chl-a signal was contaminated with suspended solid sediments and other non-biological signals, an FLH time series was incorporated into the analysis, showing a Corr. coef. of 0.54 with Chl-a. In this case, FLH also exhibited a positive relationship with TUT; the Corr. coef. was 0.27 (Fig. 11c). Negative SST anomalies were also observed during the fall and winter and at a lower frequency during the spring and summer compared to that the frequency of the upwelling response (Fig. 11d). The Corr. coef. between the SST anomalies and TUT was 0.29. From an interannual point of view, a high number of positive Chl-a and FHL anomalies was observed in 2008, 2014, and 2016 and SST anomalies were observed in 2004, 2008–2009, and 2016–2017. For the TUT time series, no interannual variability was observed, but from 2017 to the end of 2018, decreasing amounts of positive TUT were observed. The SST (Fig. 11e and g) and Chl-a images (Fig. 11f and h) obtained during the upwelling provided evidence of the oceanic response to the TUT. Along the west coast of Chiloe Island, the SST dropped by approximately 4°C and Chl-a increased ~10to 15 mg m$^{-3}$ compared with the values in the open Pacific Ocean waters. These examples demonstrated the importance of TUT in the oceanic response to wind.

### 3.3    Relationship of synoptic events with nighttime heatwaves

The long-term hourly mean of the surface air temperature (SAT) obtained from the buoy and meteorological station data showed a markedly diurnal cycle, where the SAT maximum was registered in the afternoon (15:00–18:00, LT), while the absolute minima were observed early in the morning (6:00–8:00, LT) (Fig. 12a). The histogram of the SAT absolute maxima demonstrates a bimodal structure (Fig. 12b), with an initial peak in the afternoon, as was observed in the diurnal cycle (Fig. 12a), and a second peak at night from ~21:00 to 05:00.

The balance of this subsection describes the processes involved in the SAT nighttime maximum, known in this manuscript as "nighttime heatwave events." The original time series of atmospheric pressure and zonal (wind-u) and meridional wind (wind-v) components showed the range and intensity of different variables during the same time of SAT occurrence (Fig. 12c, Fig. 12e, and Fig. 12g). This data were also obtained from the buoy and meteorological station. Alternatively, Figs. 12d, 12f, and 12h only represented the occurrence of the variable during the SAT nighttime maximum. During this time, the atmospheric pressure ranged from 990 mbar to ~1020 mbar, highlighting the presence of LAP systems (Fig. 12d), while the surface wind was predominant from the northwest and northeast direction (Fig. 12f and Fig. 12h).The time series registered 162 nighttime heatwave events from 2011 to 2017 (Fig. 13a and Fig. 13c). On average, approximately 32 events occurred every year (averaged using the complete years of 2012, 2015, and 2016).

The normalized time series of the SAT nighttime maximum (nighttime heatwave events) with the atmospheric pressure demonstrated a notable agreement (Fig. 13a), showing a high correlation coefficient (0.96) between the

variables (Fig. 13b). A similar pattern was observed between the meridional wind component and the nighttime heatwave events (Fig. 13c), also registering a high correlation coefficient (0.78) (Fig. 13d). The temperature range from these events was 5 to 20°C, with the most common temperatures between 10 and 12°C (Fig. 13e). The monthly histogram of the nighttime heatwave events showed most cases occurred in the fall and winter, with fewer incidences in the summer (Fig. 13f). Figure 14 presents one example of the 162 events detected in this study, which occurred during fall 2011, as shown in the atmospheric data from the oceanographic buoy installed in Puyuhuapi Fjord. The maximum SAT was observed on 21 April2011 at midnight (00:00, LT), coinciding with a decreased atmospheric pressure and increased surface-wind intensity (Fig. 14a).

The ERA5 reanalysis climate data sets were used to explore the causes of the augmented air temperature (Fig. 14b–14g). Images of surface wind and atmospheric pressure from before the event showed the predominance of a westerly wind from 45°–56° S and northerlies from 30°–35° S (Fig. 14b). At the same time, the SAT showed a meridional gradient, in which the high air temperature covered the northern domain of the image (30°–40° S) (Fig. 14c). At midnight on 21 July, 2011 (00:00, LT), a LAP system arrived in the eastern Austral Pacific Ocean and moved northward, interacting with the southern edge of an SPSA system. LAP systems rotate clockwise with intense winds of ~25 m s$^{-1}$ and a minimum atmospheric pressure of 958 mbar (Fig. 14d), and the west and northwest winds from the LAP transported the warm air from the area with the maximum air temperature. The latter was located north of 40°S, advected the maximum air temperature southward, and contributed to the increased air temperature in Patagonia, as shown in Fig. 14a. High air temperatures reached the southern part of Patagonia, close to the Magellan Strait, due to the LAP winds (Fig. 14e). Atmospheric conditions returned to normal days after the passage of the LAP (Fig. 14f and Fig. 14g) as shown in Fig.14b and 14c.

A second example, using atmospheric data from the winter of 2012, better demonstrated the increased SAT over Patagonia due to the LAP system influence (Fig. 15). In this case, the maximum air temperature was again registered when the intensity of the wind had increased, and atmospheric pressure had been low (Fig. 15a). Before this event, less intense winds were from the north and northwest, and the high air temperature presented the usual meridional gradient (Fig. 15b and Fig. 15c). At midnight of 18 July 2012, a LAP system entered the study area and advected high air temperature from the subtropical area southward to Patagonia. During this nighttime heatwave event, warm air was transported along the coast of Patagonia to ~56° S (Fig. 15d and Fig. 15e). Pre-event atmospheric conditions were restored one day after the passage of the LAP system (Fig. 15f and Fig. 15g).Other studies were incorporated in the manuscript to demonstrate the relationship between atmospheric pressure, winds, and surface air temperature during nighttime heatwaves events (see supplementary material, Fig. S9).

4.     **Discussion**

The combination of QuikSCAT, ASCAT, and ERA5surface-wind products, together with in situ measurements of wind from oceanographic buoys and meteorological stations, has facilitated the understanding of the surface-wind variability in the eastern Austral Pacific Ocean and the Patagonian interior. Surface winds were generally westerlies (Fig. 2), and the synoptic scale dominated wind variability due to the influence of the low/high atmospheric pressure

systems with winds from the northerly /southerly directions, respectively (Fig. 3-6). Implications of the synoptic scale events on the atmosphere-ocean interaction is the focus of this section of the manuscript, owing to the importance of the winds to the oceanic responses, such as ET and EP, and their influence on the Patagonian climate.

### 4.1 Surface-wind variability

Satellite data on the long-term surface-wind daily means, over the period 1999–2015, demonstrated that between 42°–45° S, the normal wind was perpendicular to the coast and blew from the west. From 45°–56° S, the predominant wind direction changed to the northwest, reaching its highest intensity in the Magellan region where it blew parallel to the coast. At the other end of the study region (40°–42° S), the predominant wind was from the southwest, although the intensity was less than in the Magellan region (Fig. 2). To date, the wind regime for this region has only been presented as a conceptual model to show the influence of the westerlies on the westerly drift current (Thiel et al., 2007; Arkhipkin et al., 2009; KilianandLamy, 2012) and to present the general atmospheric circulation applicable to the west coast of South America (Rahn and Garreaud, 2013; Talley et al., 2011). Even though maps similar to Fig. 2 were presented in Aguirre et al. (2012) and Saldías et al. (2018) using QuikSCAT data, details of the surface-wind behavior could not be determined. In addition, winds regime studies, which included derived variables such as EP and ET, focused on the central and northern region of the Chilean and Peruvian coasts, north of40° S. The main goals of these studies were to explain the dynamic of the SPSA and improve the understanding of the wind's influence on this circulation regime (Ancapichun and Garcés-Vargas, 2015; Bravo et al., 2016; Fuenzalida et al., 2008; Schneider et al., 2017). Recently, the behavior and evolution of the ET in northern Patagonia and its implications in ocean response was investigated (Narváez et al., 2019). In the next section, these results will be incorporated and discussed.

LAP and HAP systems dominated mode 1 of the EOF, contributing ~30 % of the total variance (Fig. 3–6). In this mode, southerlies related to the passage of the HAP systems, and northerlies produced by the LAP systems (Fig. 7), occurred in a time scale of 16.5 days (Fig. 4–6 and Fig. 8). This illustrated the variability of surface winds in the eastern Austral Pacific Ocean, complementing the westerly winds which have been seen to dominate the wind regime in average and seasonal data (Fig. 2).

EOF analysis detected wind from the west in mode 2, accounting for 22% of the total variance. This wind occurred mainly during spring-summer before veering to an easterly wind for fall-winter (Fig. 3–6). The cycle change was observed in this mode using the individual QuikSCAT and ASCAT data sets and confirmed by the continuous data set of the ERA5 reanalysis (Fig. 8). During the first period, an annual cycle dominated mode 2 (1999–2009), but in the second period (2009–2015), this dominance reduced, and cycle periods of 27.5 days and 16.5 days were observed (Fig. 8). The period of 16.5 days denoted the importance of the synoptic time scale, while the 27.5-day cycle suggested the influence of the recently reported, Southern Hemisphere´s baroclinic annular mode (BAM),which has been described as displaying an energy band lasting between 20 and 30 days (Thompson and Barnes, 2014; Thompson and Woodworth, 2014). The BAM's influence was observed by Ross et al., (2015) in a Patagonian fjord (47.8° S), using Acoustic Doppler current profiler(ADCP) data combined with in situ surface-wind

and atmospheric pressure records. These data highlighted the contribution of this atmospheric phenomenon to the intensification and frequency of the LAP systems that occur throughout the Patagonian. In addition, Narváez et al. (2019) reported the dominance of the BAM on an intra-seasonal time scale, showing the essential influence of this cycle on the atmospheric and oceanographic conditions of northern Patagonia (40°–45°S).

Finally, the EOF analysis allowed for the detection of the high surface-wind variability in the eastern Austral Pacific Ocean, showing the dominance of the atmospheric pressure systems (e.g., LAP and HAP systems) over the various time scales. These atmospheric synoptic events occurred throughout the study region, coinciding with the significant areas of the strongest westerly wind belt on earth (Chelton et al., 2004).

### 4.2    Atmospheric-ocean interactions

The long-term ET mean showed that this movement ran parallel to the coast from 40°–48° S. Then, from 48° S to 58° S, it ran perpendicular (onshore) to the coastline, showing a higher magnitude in the Magellan region (Fig. 9). Studies have shown that, when onshore ET occurred, downwelling conditions prevailed and particulates were transported to the coast (Stewart, 2002), favoring the retention of eggs and larvae in the coastal zone (Epifanio and Garvine, 2001; Garland et al., 2002). It has also been shown that, when offshore ET occurred, upwelling processes dominated along the coastline, favoring primary biologial production (Escribano et al., 2016; Iriarte et al., 2012; Montero et al., 2007). As argued in the previous section, synoptic-scale atmospheric events, such as LAP and HAP systems, dominated wind variability within the study area, especially in its northern domain, where the southern edge of the SPSA arrived during spring-summer. During this time of the year, southerly winds influenced the region, producing offshore ET, as shown by the ET time series for the northern (42.7° S) part of the study area (Fig. 10). Then, the upwelling process occurred along the coastline of Chiloe Island, as was demonstrated by the increased Chl-a and the drop of SST in this area (Fig. 11). Also, EP velocity was positively helping the upward movement of oceanic water, which enhanced the injection of nutrients into the surface layer of the water (Rykaczewski and Checkley, 2008). Quantification of the Ekman downwelling/upwelling processes and their impact on the ocean response demonstrated the contributions of the EP and the ET to the TUT in northern Patagonia (along with the west coast of Chiloé Island).These results imply that wind stress curl and offshore ET play essential roles in the upward displacement of rich oceanic water to the surface layer. During the annual cycle, favorable upwelling conditions were observed from November to April (Figs. 10 and 11), the time of year with more intense photosynthetically available radiation (PAR) for phytoplankton species (Daneri et al., 2012).To date, coastal upwelling quantification, using only the ET, has been reported as far south as the central coastal region of Chile (~36° S) (Sobarzo and Djurfeldt, 2004; Sobarzo et al., 2007), and recent analyses have been extended to 45°S (Narváez et al., 2019). However, our work has shown that coastal upwelling can also occur by the contribution of EP and must be added to the TUT quantification for a realistic evaluation of the ocean response to surface winds. For example, in the California upwelling system, EP is more significant than ET for the TUT, especially during spring and summer (Pickett and Paduan, 2003). In northern Chile (27°–32°S), EP represented ~40% of the TUT, causing changes in the

SST spatial structure (Bravo et al., 2016). Around Cabo Frío (22°S/ 42°W), EP was also the primary contributing process in the upwelling of the coldest water to the surface layer (Castelao and Barth, 2006).

From an interannual point of view, the TUT favored upwelling from spring 2015 to summer 2016, contributing to high Chl-a and FLH readings during summer 2016 (Fig. 11). A strong harmful algal bloom (HAB) was reported in northern Patagonia during February and March 2016, causing the death of 40.000 t of salmon (Díaz et al., 2019; Paredes et al., 2019). The main factors involved in the 2016 HAB included increased solar radiation, SST, and water column stratification, which were highlighted as trigger mechanisms (Léon-Muñoz et al., 2018). However, the

results presented in this manuscript show that EP and ET favor the upwelling of nutrient-rich water to the euphotic layer, which can contribute to HAB development. Additionally, a high ammonium concentration was observed two months later in the open oceanic water off the west coast of Chiloé Island (41°46'15" S / 75°43'31" W) due to the shedding of 4.600 t of dead salmon to the sea (Buschmann et al., 2016). As Fig. 11 shows, during this time, EP favored the vertical ascent of water, inhibiting the sinking of the biochemical waste. Therefore, the Ekman

upwelling process must be something for the decision-makers to consider during future environmental emergencies.

In general, however, downwelling conditions dominated by onshore ET, were observed in the study area, especially in the south close to the Magellanic region (Fig. 9 and Fig. 10). We have hypothesized that the irregular orographic structure of the coastline from 44°S–56° S, where the coast is comprised of many islands and channels, could reduce the possibility of the oceanic water sinking at the coastline passing into the interior of the Patagonian fjords and

carrying the eggs and larvae of many species and nutrients and enhancing biological production.

It was not only the ocean that responded to the synoptic-scale variability of the surface wind; the atmospheric conditions were also influenced. This study registered approximately 160 events in which the SAT nighttime maximum ("nighttime heatwave events") occurred in response to the influence of low atmospheric pressure systems with winds from the northwest and northeast directions predominating, registering a high correlation coefficient

between the SAT nighttime maximum with the atmospheric pressure and meridional wind components (Fig. 12 and Fig. 13). Various examples demonstrated the importance of the synoptic-scale events in modifying climate conditions in the Austral region (Fig. 14,Fig. 15 and Fig. S9), where the LAP systems contribute to the origin of the nighttime heatwave events.

A conceptual model was built to explain the source of the nighttime heatwave events (Fig. 16). In this model, two

atmospheric pressure systems participated: a permanent high pressure located in the north (SPSA), which transported warm air from the subtropical region (over the 40° S), and a synoptic LAP system, which originated in the south, with cold air from the Polar zone (Fig. 16a). The LAP originated in the Austral-Pacific Ocean, and the system moved northward with intense winds rotating clockwise. The northward-moving LAP stopped when it encountered the southern edge of the SPSA, at approximately 40° S (Fig. 16b). Then, the stronger west and

northwest winds from the LAP pulled in the warm air from the SPSA and advected its heat southward to Patagonia. These events occur more frequently at nighttime, and their impact on the Patagonian climate depends on the intensity of the LAP system winds and the heat content of the SPSA. In the contexts of climate change and variability, any changing trends in these events should be considered as mechanisms that could contribute to increased glacial meltwater and alteration of the Austral climate.

## 5.    Conclusions

In our study, satellite and reanalysis wind data were used to understand surface-wind variability in the eastern Austral Pacific Ocean, a region generally dominated by strong westerlies, and the SPSA. The EOF demonstrated that within the area, modes 1, 2, and 3 of wind variability showed synoptic time scale dominance due to the effects of low and high atmospheric pressure systems. Generally, downwelling conditions prevailed in the study region due to onshore ET, but offshore ET and upward EP were observed during spring and summer in the northern domain (~40°–48° S), contributing to reduced SST and increased Chl-a. The arrival of the southern edge of the SPSA during spring and summer created upwelling conditions dominated by EP; this is the first time that this condition has been reported so far south. In addition, the SPSA was involved in generating the nighttime heatwaves, acting with the LAP systems to produce the night-time air temperature maxima.

**Data availability**. All data sets used in this manuscript can be requested from the corresponding author.

**Supplement**. The supplement related to this article is available online.

**Author contributions**. IPS: designed the experiment, collection, and analysis of the satellite data, and was the manuscript leader. RS: collection and analysis of the satellite and in situ data, as well as manuscript revision. WS: designed the experiment, collection, and analysis of the satellite data, as well as manuscript revision. PL: data analysis of ERA5 and generation of Figure 1. DD: data analysis of ERA5 and validation process. EN: data analysis. CAC: validation process. GD: manuscript revision. All authors contributed to the writing this manuscript.

**Competing interest**. The authors declare that they have no conflict of interest.

## Acknowledgments

Surface-wind data were collected as part of FONDECYT Grants 3120038, and 11140161, by Dr. Iván Pérez-Santos, with assistance from Dr. Wolfgang Schneider's research group. Financial support was also provided by Centro Copas Sur Austral PFB31 and AFB170006. We thank to Raul Montoya for the EOF function. We are grateful to Centro Copas Sur Austral for providing data from its oceanographic buoy and a partial scholarship for Romanet Seguel to complete a Magister in Oceanography at the University of Concepción, Chile. We thank Centro de Investigación en Ecosistemas de la Patagonia (CIEP) for providing meteorological information. The ERA5 reanalysis data were provided by the Copernicus Climate Change Service (C3S) (2017): ERA5: Fifth generation of the ECMWF atmospheric reanalyses of the global climate. Copernicus Climate Change Service Climate Data Store (CDS). We also thanks to the FONDEQUIP EQM160167 and to the Chilean Navy from providing wind data from oceanographic buoy and navy lighthouses. We appreciate the tremendous effort of the anonymous reviewers which led to improved manuscript quality.

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

**Figure captions**

Figure 1. Map of the study area and geographical position of the sampling stations. Bathymetric image of the seafloor and topography obtained from https://www.gmrt.org/GMRTMapTool.

Figure 2. Long-term mean of daily surface wind from (**a**) QuikSCAT (1999–2009), (**b**) ASCAT (2007–2016) and (**c**) ERA5 reanalysis climate data sets (1999–2015). Black line: standard deviations of daily data. Colored bar: surface-wind magnitude.

Figure 3. Eastern Austral Pacific Ocean, 1999 to 2015: normalized eigenvector patterns, from QuikSCAT (**a**, **b**, and **c**), ASCAT (**d**, **e**, and **f**) and ERA5 reanalysis (**g**, **h** and **i**).

Figure 4. (**a**, **d**, and **g**) Normalized time series of the time-dependent coefficient (black lines) from the 30-day, low pass filtered time series (solid red lines). (**b**, **e**, and **h**) Global wavelet spectra (black solid lines) with 95% confidence interval (red dashed lines), and (**c**, **f**, and **i**) long-term monthly mean of EOF modes from surface wind daily data on QuikSCAT from 1999 to 2009: mode 1 (**a**, **b**, and **c**), mode 2 (**d**, **e**, and **f**), mode 3 (**g**, **h**, and **i**).

Figure 5. (**a**, **d**, and **g**) Normalized time series of the time-dependent coefficient (black lines) from the 30-day, low pass filtered time series (solid red lines). (**b**, **e**, and **h**) Global wavelet spectra (black solid lines) with 95% confidence interval (red dashed lines) and (**c**, **f**, and **i**) long-term, monthly mean of EOF modes from surface wind daily data on ASCAT from 2007 to 2015: mode 1 (**a**, **b**, and **c**), mode 2 (**d**, **e**, and **f**), mode 3 (**g**, **h**, and **i**).

Figure 6. (**a**, **d**, and **g**) Normalized time series of the time-dependent coefficient (black lines) from the 30-day, low pass filtered time series (solid red lines). (**b**, **e**, and **h**) Global wavelet spectra (black solid lines) with 95% confidence interval (red dashed lines), and (**c**, **f**, and **i**) long-term monthly mean of EOF modes from surface wind daily data of ERA5 reanalysis from 1999 to 2015: mode 1 (**a**, **b**, and **c**), mode 2 (**d**, **e**, and **f**), mode 3 (**g**, **h**, and **i**).

Figure 7. Snapshots of the surface winds representing EOF eigenvector spatial structures for mode 1 (**a** and **b**), mode 2 (**c** and **d**); and mode 3 (**e** and **f**). Surface wind and atmospheric pressure data were obtained from the ERA5 reanalysis climate product. The surface-wind vectors were plotted with a spatial resolution of 1°×1°. The red rectangle in (**a**) indicates the study area.

Figure 8. Morlet wavelet power spectrum applied to the three series of the EOF time-dependent coefficient from QuikSCAT (**a**, **c**, and **e**), ASCAT (**b**, **d**, and **f**), and ERA5 (**g**, **h**, and **i**).The fine contour lines enclose regions of confidence levels of >95% for a red noise process with a lag 1 coefficient between 0.52 and 0.55, and the thick contour lines indicate the cone of influence. The color bar relates colors on the power spectrum.

Figure 9. The long-term mean of daily ET (red arrows), and EP (color bars) from (**a**) QuikSCAT (1999–2009), (**b**) ASCAT (2007–2016), and (**c**) ERA5 reanalysis (1999–2015). The black lines represent the zero value of EP, where a positive number is a region favorable to upwelling and negative to downwelling.

Figure 10. Quantification of the cross-shore transport using ERA5 reanalysis from the north, center, and south time series (see Fig. 1 for the position) from 1999–2018 (**a**, **d**, and **g**) representing the long-term daily mean, (**b**, **e**, and **h**) the long-term monthly mean, and (**c**, **f**, and **i**) cumulative ET, EP, and TUT. The TUT is the sum of the ET and EP. The positive/negative values of transport indicate upwelling/downwelling conditions.

Figure 11. Time series of (**a**) the TUT from ERA5, (**b**) the Chl-a anomalies, (**c**) the FLH anomalies and (**d**) the SST anomalies from the MODIS-Aqua satellite data. (**e-h**) Examples showing the ocean response to ET and EP along the northern coast of Patagonia.  SST (**e** and **g**) and Chl-a (**f** and **h**) from MODIS-Aqua. Time series of TUT (**a**) was obtained from point north of Chiloé Island (see Fig. 10 **a**) and time series from (**b**) Chl-a, (**c**) the FLH, and (**d**) the SST anomalies were extracted from the point closer to the position of TUT time series (solid black square in Fig. 11**e**).

Figure 12. (**a**) SAT long-term hourly means with (**b**) histogram of the maximum SAT. The red shaded area in (**b**) shows the time of the second air temperature maxima. The error bars in (**a**) represent the standard deviations of SAT. (**c**, **e**, and **g**) Complete data set of atmospheric pressure and zonal (wind-u) and meridional wind (wind-v) components. (**d**, **f**, and **h**) Atmospheric pressure and zonal and meridional wind values related to the second SAT maxima. Data were obtained from the Puyuhuapi Fjord oceanographic buoy and meteorological station in the period 2011−2017.

Figure 13. Time series of the nighttime heatwave events. (**a**) Normalized time series of SAT nighttime maximum (black dots) and atmospheric pressure (red dots).  (**b**) Cross-correlation coefficient between variables from (**a**). (**c**) Normalized time series of SAT nighttime maximum (black dots) and meridional wind component (red dots). (**d**) Cross-correlation coefficient between variables from (**c**). (**e**) Histogram and (**f**) long-term monthly mean from time series of SAT nighttime maximum in the period 2011 to 2017. Data were obtained from the Puyuhuapi Fjord oceanographic buoy (2011–2013) and meteorological station (2014–2017). From July 2013 to April 2014, no data were collected. The blue circle in (**a** and **c**) denotes the position of the nighttime heatwave events described in Figs. 14 and 15.

Figure 14. Hourly air temperature, atmospheric pressure, and wind speed data from the Puyuhuapi Fjord oceanographic buoy (**a**) and surface winds, atmospheric pressure and surface air temperature from the ERA5 reanalysis climate product (**b**–**g**), during April 2011. The surface-wind vectors (**b**, **d**, and **f**) were plotted with a spatial resolution of 1°×1°.

Figure 15. Hourly air temperature, atmospheric pressure, and wind speed data from the Puyuhuapi Fjord oceanographic buoy (**a**) and surface winds, atmospheric pressure and surface air temperature from the ERA5 reanalysis climate product (**b**–**g**), during July 2012. The surface-wind vectors (**b**, **d**, and **f**) were plotted with a spatial resolution of 1°×1°.

Figure 16. A conceptual model of the "Night-time heatwave event" in the Eastern Austral Pacific Ocean. (**a**) The initial condition, where a low atmospheric pressure system with cold air and a high atmospheric pressure system with warm air are regionally present, although separate; (**b**) the low atmospheric pressure system moves northward and encounters the high atmospheric pressure system, transporting warm air to Patagonia.

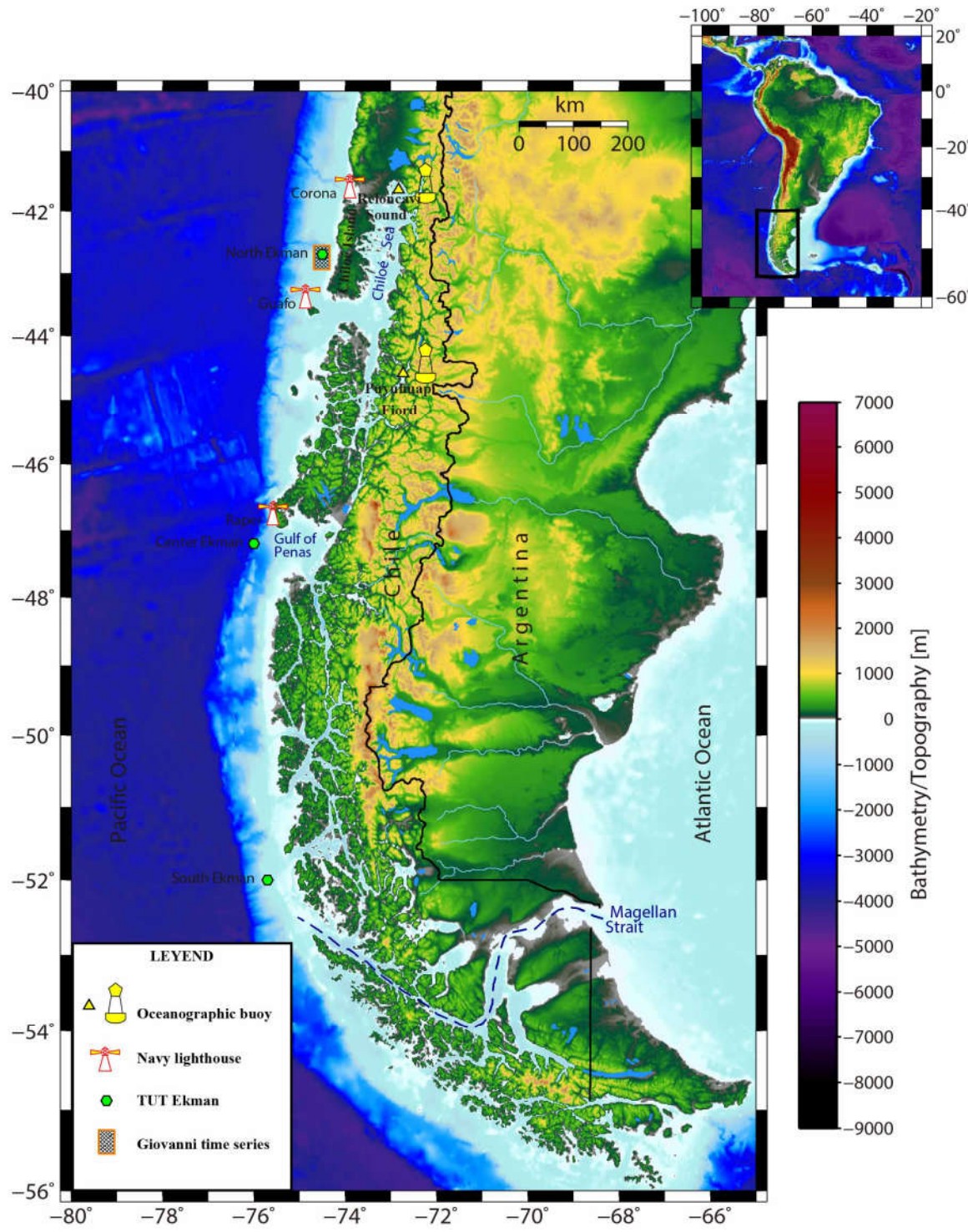

Figure 1. Map of the study area and geographical position of the sampling stations. Bathymetric image of the seafloor and topography obtained from https://www.gmrt.org/GMRTMapTool.

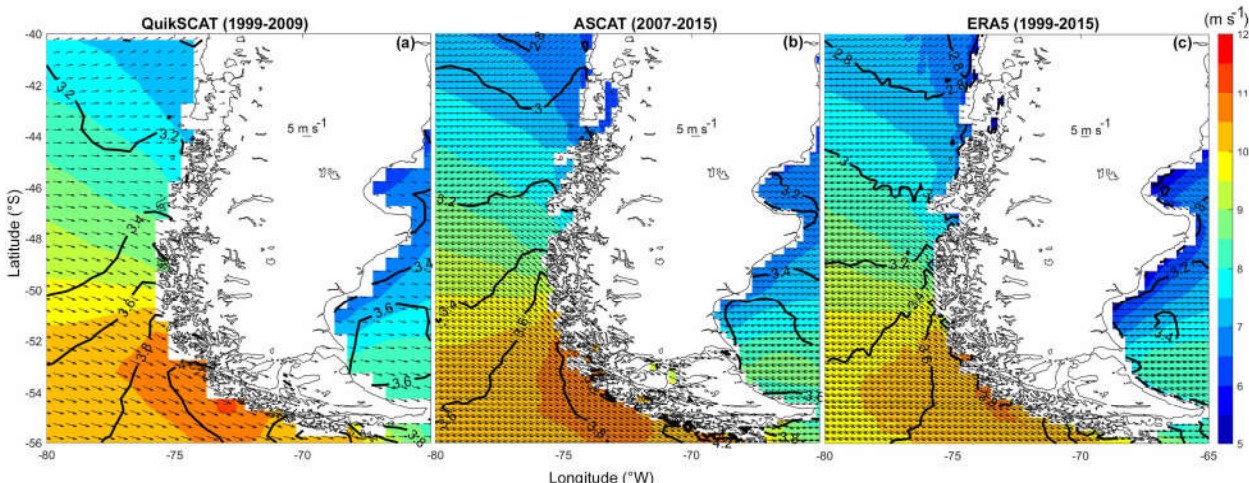

Figure 2. Long-term mean of daily surface wind from (**a**) QuikSCAT (1999–2009), (**b**) ASCAT (2007–2016) and (**c**) ERA5 reanalysis climate data sets (1999–2015). Black line: standard deviations of daily data. Colored bar: surface-wind magnitude.

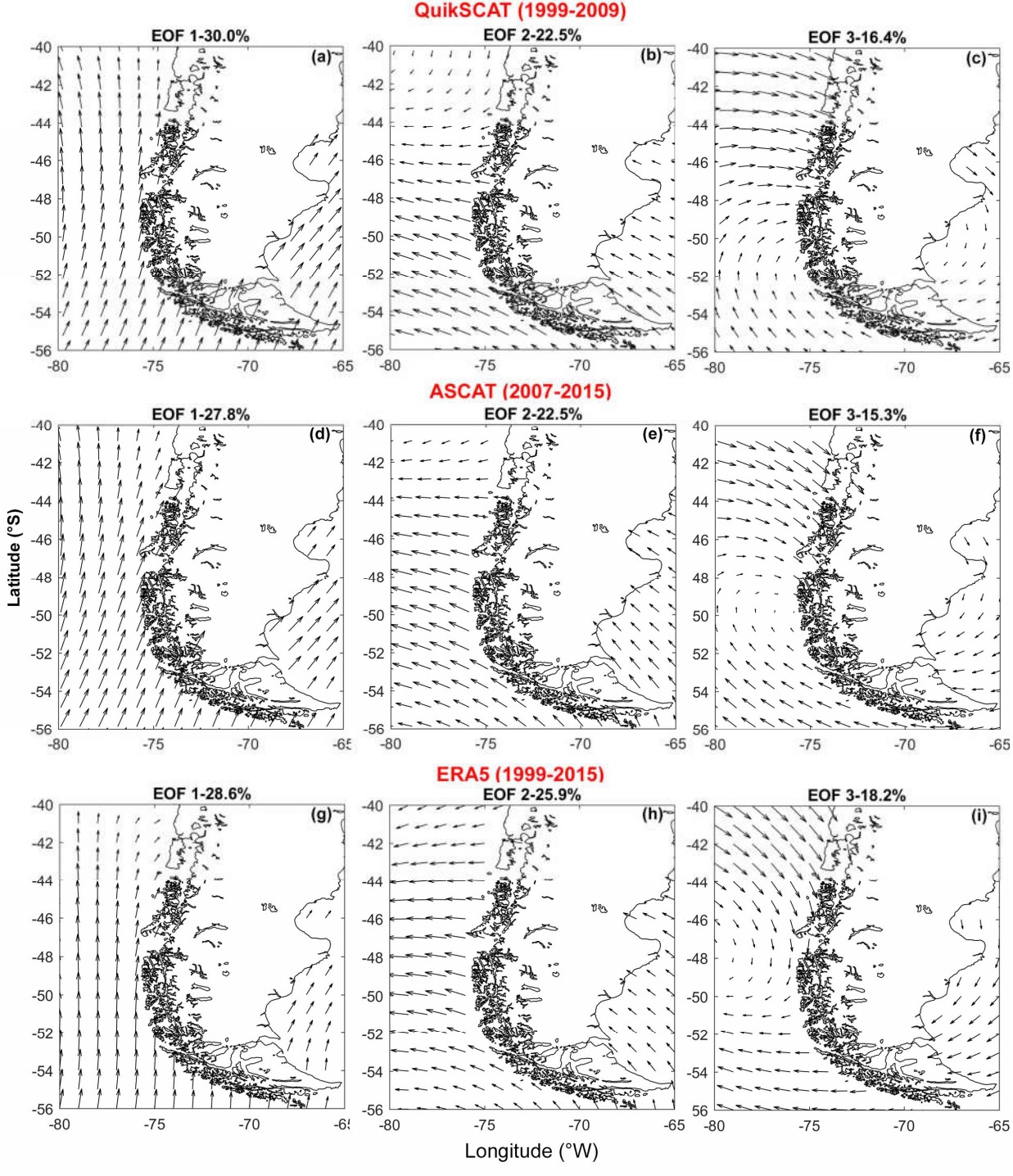

Figure 3. Eastern Austral Pacific Ocean, 1999 to 2015: normalized eigenvector patterns, from QuikSCAT (**a**, **b**, and **c**), ASCAT (**d**, **e**, and **f**) and ERA5 reanalysis (**g**, **h** and **i**).

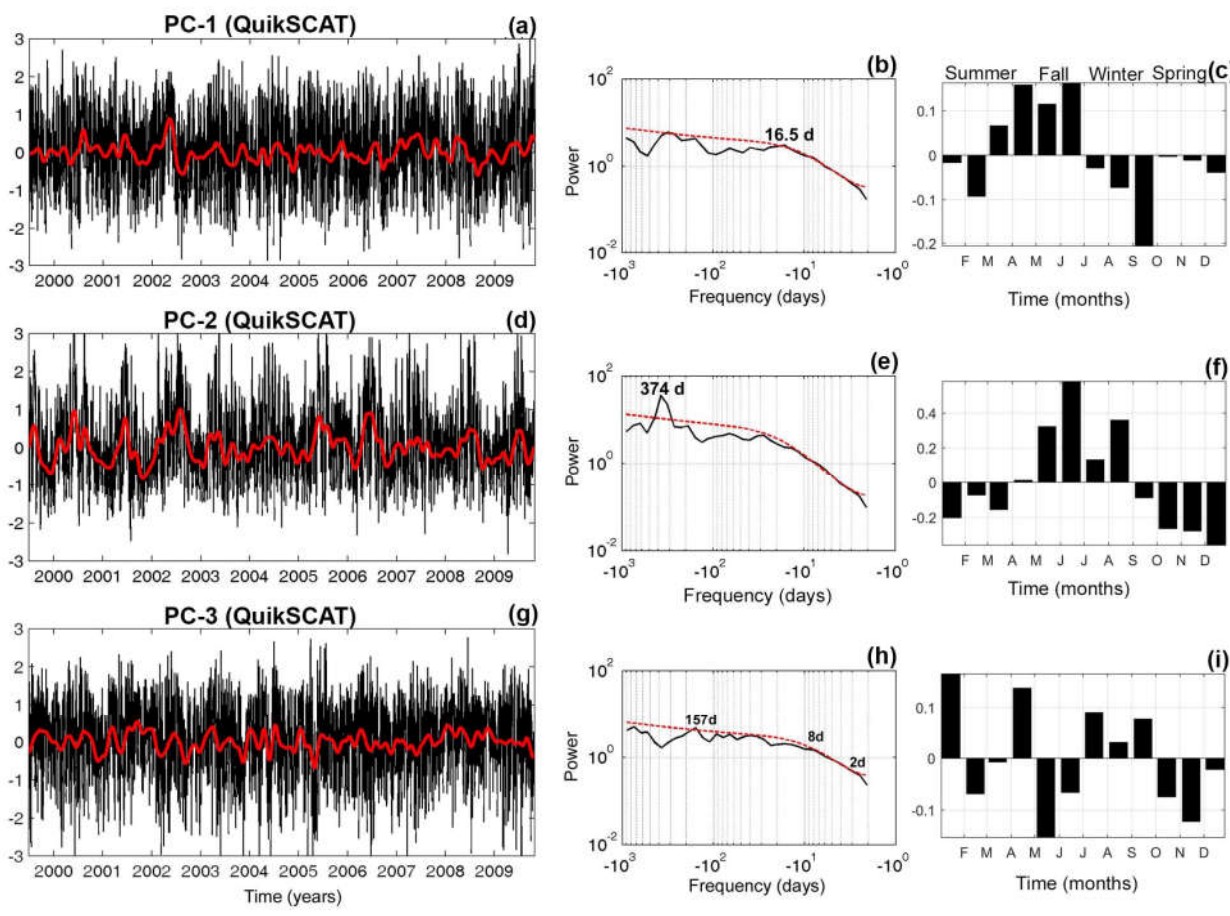

Figure 4. (**a**, **d**, and **g**) Normalized time series of the time-dependent coefficient (black lines) from the 30-day, low pass filtered time series (solid red lines). (**b**, **e**, and **h**) Global wavelet spectra (black solid lines) with 95% confidence interval (red dashed lines), and (**c**, **f**, and **i**) long-term monthly mean of EOF modes from surface wind daily data on QuikSCAT from 1999 to 2009: mode 1 (**a**, **b**, and **c**), mode 2 (**d**, **e**, and **f**), mode 3 (**g**, **h**, and **i**).

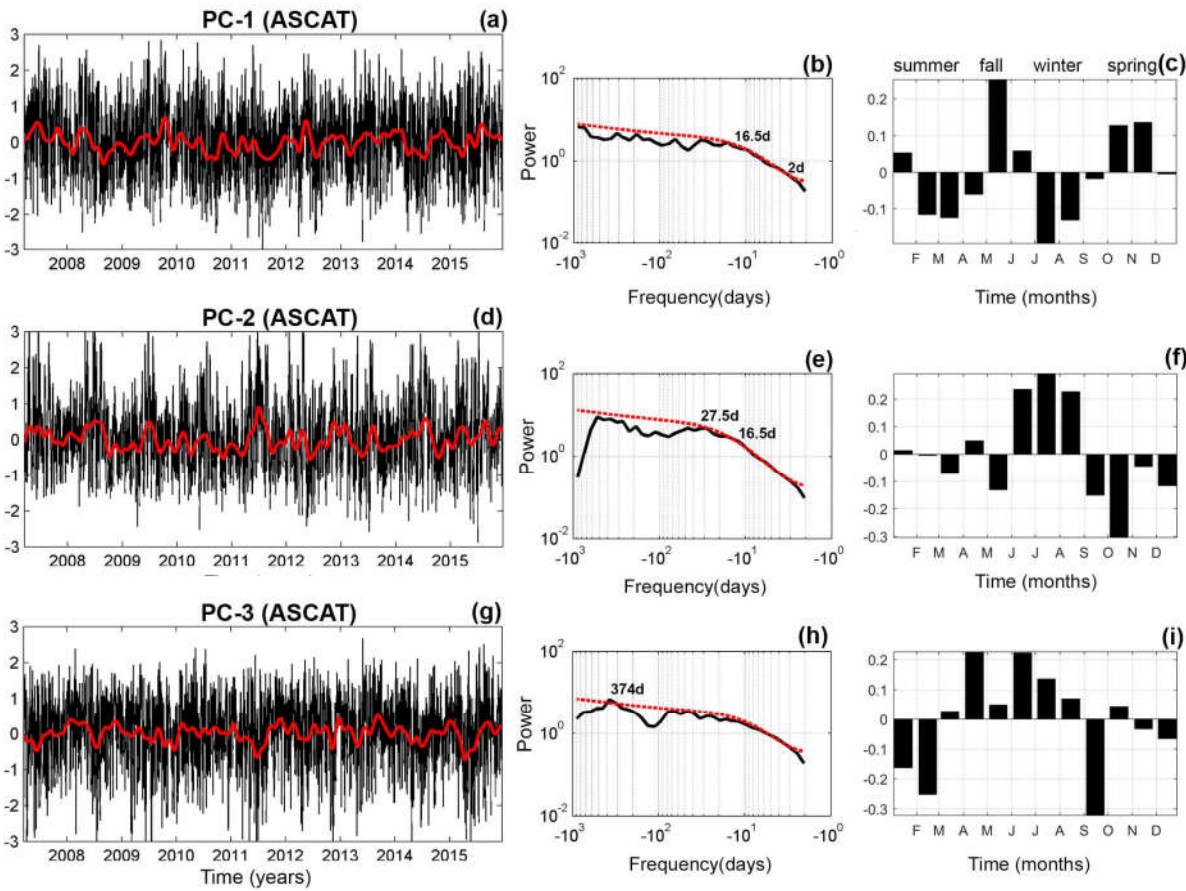

Figure 5. (**a**, **d**, and **g**) Normalized time series of the time-dependent coefficient (black lines) from the 30-day, low pass filtered time series (solid red lines). (**b**, **e**, and **h**) Global wavelet spectra (black solid lines) with 95% confidence interval (red dashed lines) and (**c**, **f**, and **i**) long-term, monthly mean of EOF modes from surface wind daily data on ASCAT from 2007 to 2015: mode 1 (**a**, **b**, and **c**), mode 2 (**d**, **e**, and **f**), mode 3 (**g**, **h**, and **i**).

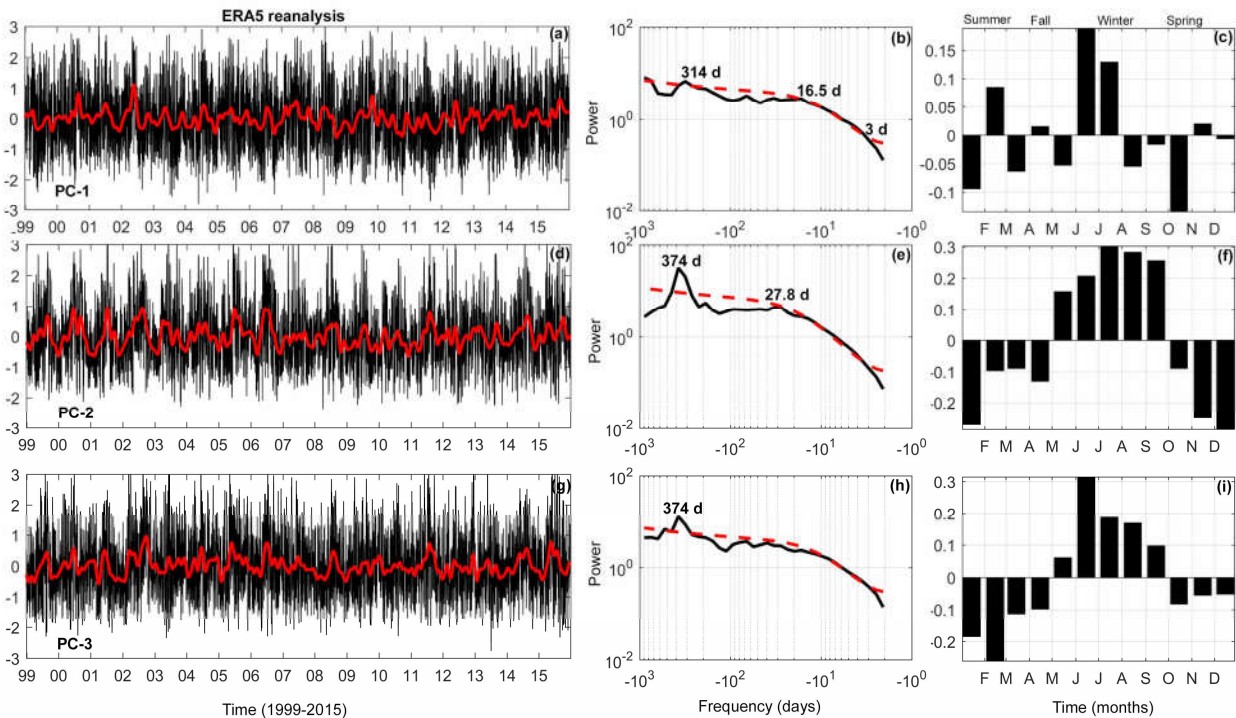

Figure 6. (**a**, **d**, and **g**) Normalized time series of the time-dependent coefficient (black lines) from the 30-day, low pass filtered time series (solid red lines). (**b**, **e**, and **h**) Global wavelet spectra (black solid lines) with 95% confidence interval (red dashed lines), and (**c**, **f**, and **i**) long-term monthly mean of EOF modes from surface wind daily data of ERA5 reanalysis from 1999 to 2015: mode 1 (**a**, **b**, and **c**), mode 2 (**d**, **e**, and **f**), mode 3 (**g**, **h**, and **i**).

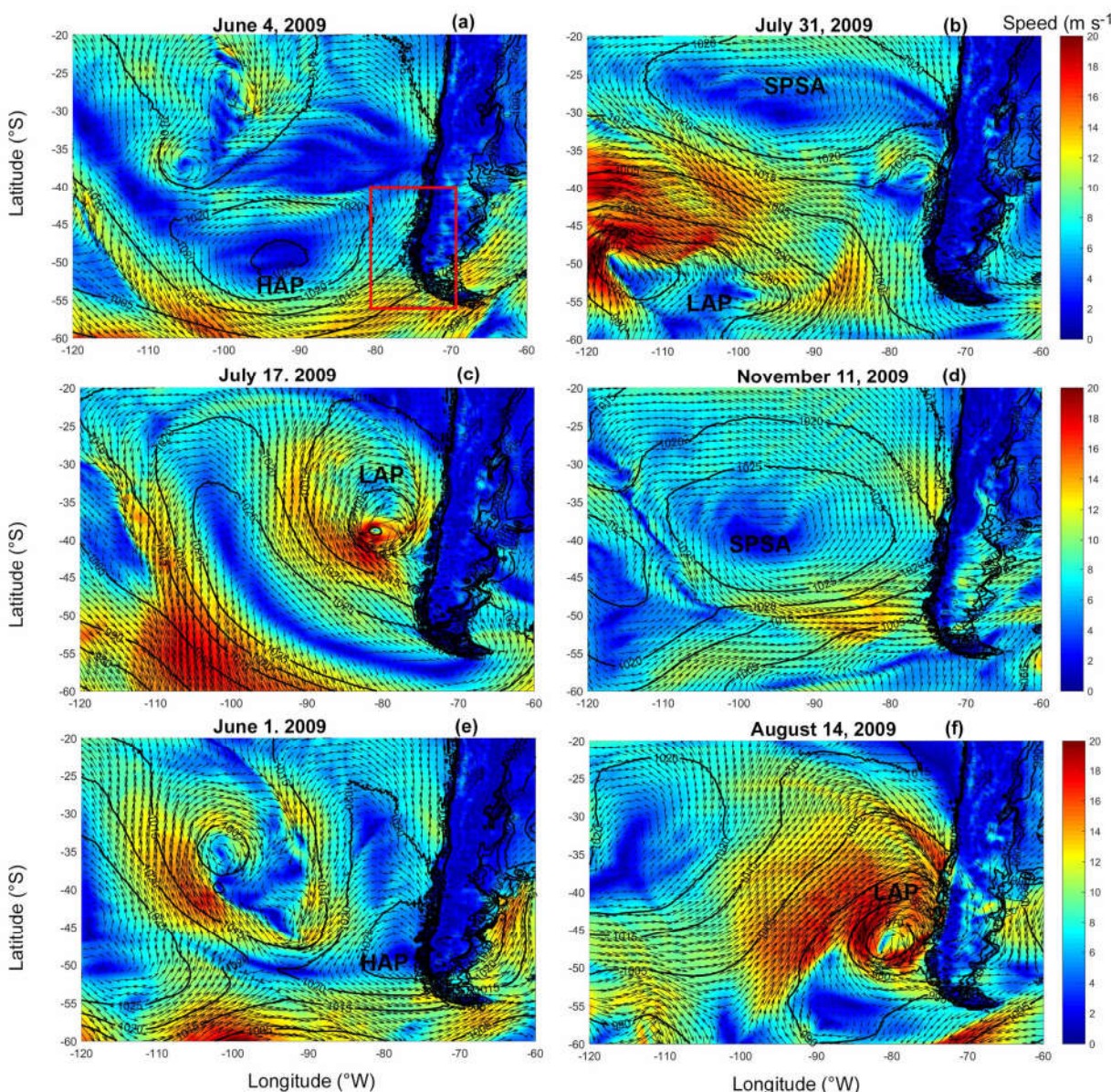

Figure 7. Snapshots of the surface winds representing EOF eigenvector spatial structures for mode 1 (**a** and **b**), mode 2 (**c** and **d**); and mode 3 (**e** and **f**). Surface wind and atmospheric pressure data were obtained from the ERA5 reanalysis climate product. The surface-wind vectors were plotted with a spatial resolution of 1°×1°. The red rectangle in (**a**) indicates the study area.

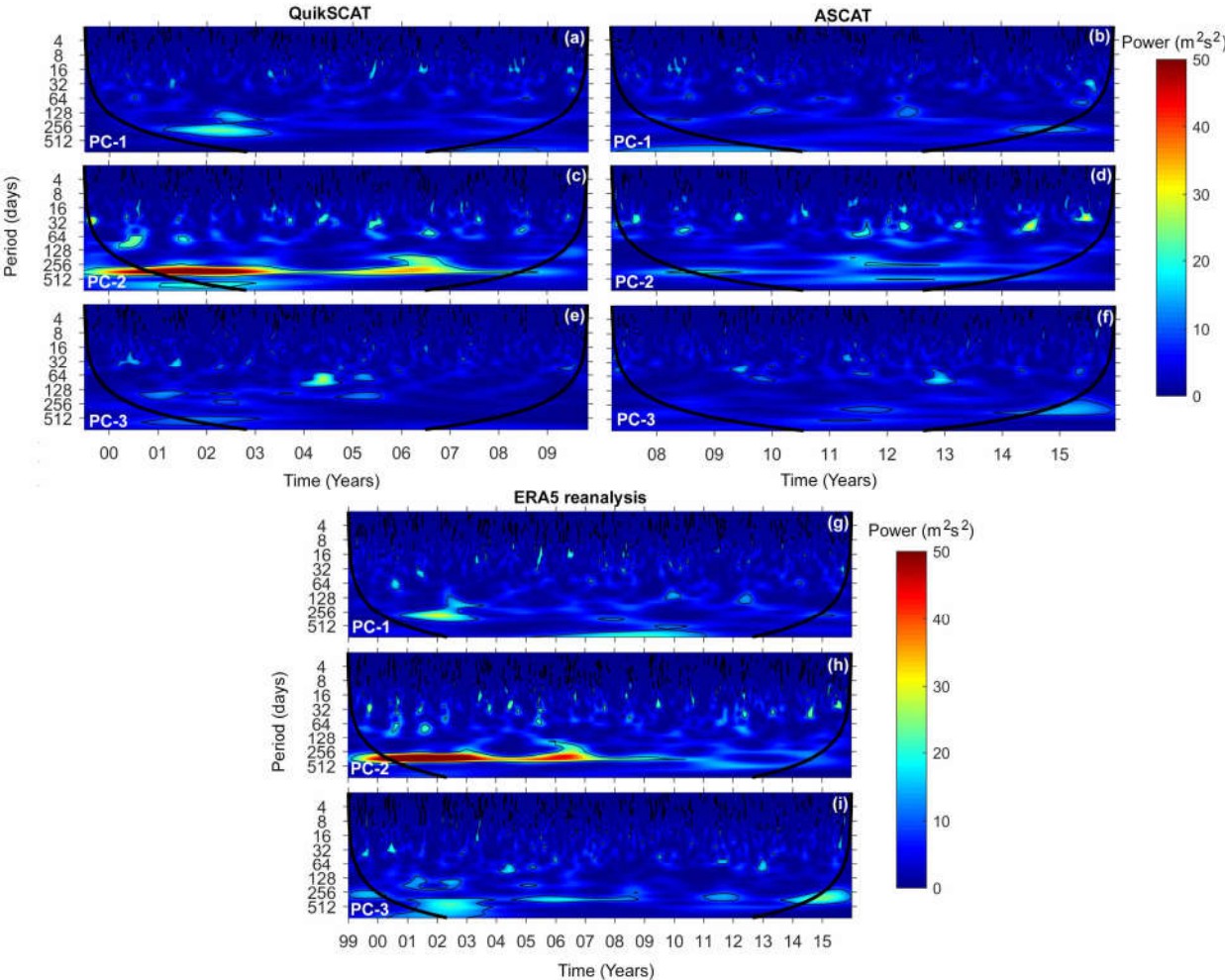

Figure 8. Morlet wavelet power spectrum applied to the three series of the EOF time-dependent coefficient from QuikSCAT (**a**, **c**, and **e**), ASCAT (**b**, **d**, and **f**), and ERA5 (**g**, **h**, and **i**).The fine contour lines enclose regions of confidence levels of >95% for a red noise process with a lag 1 coefficient between 0.52 and 0.55, and the thick contour lines indicate the cone of influence. The color bar relates colors on the power spectrum.

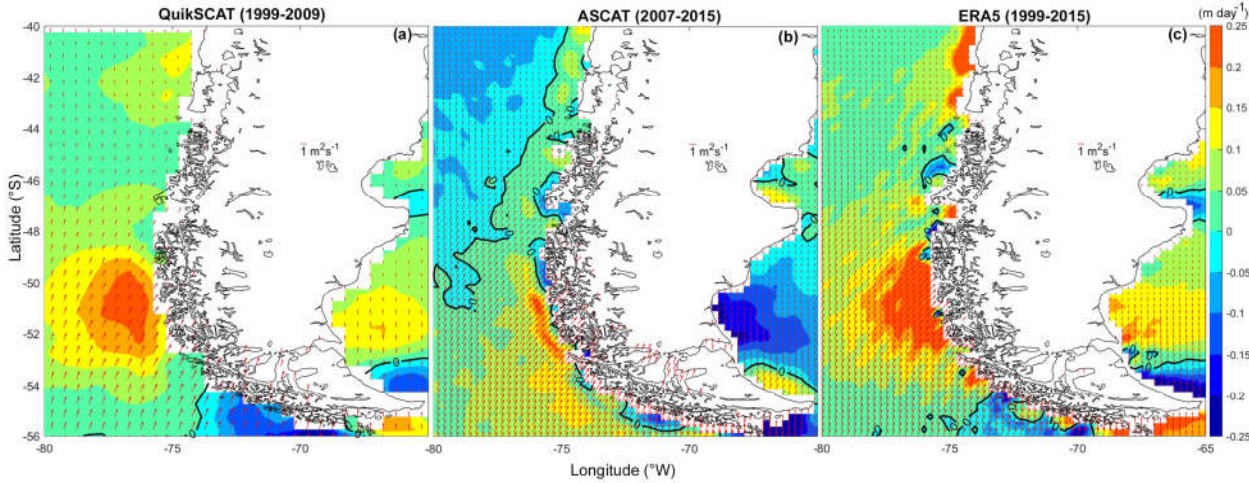

Figure 9. The long-term mean of daily ET (red arrows), and EP (color bars) from (**a**) QuikSCAT (1999–2009), (**b**) ASCAT (2007–2016), and (**c**) ERA5 reanalysis (1999–2015). The black lines represent the zero value of EP, where a positive number is a region favorable to upwelling and negative to downwelling.

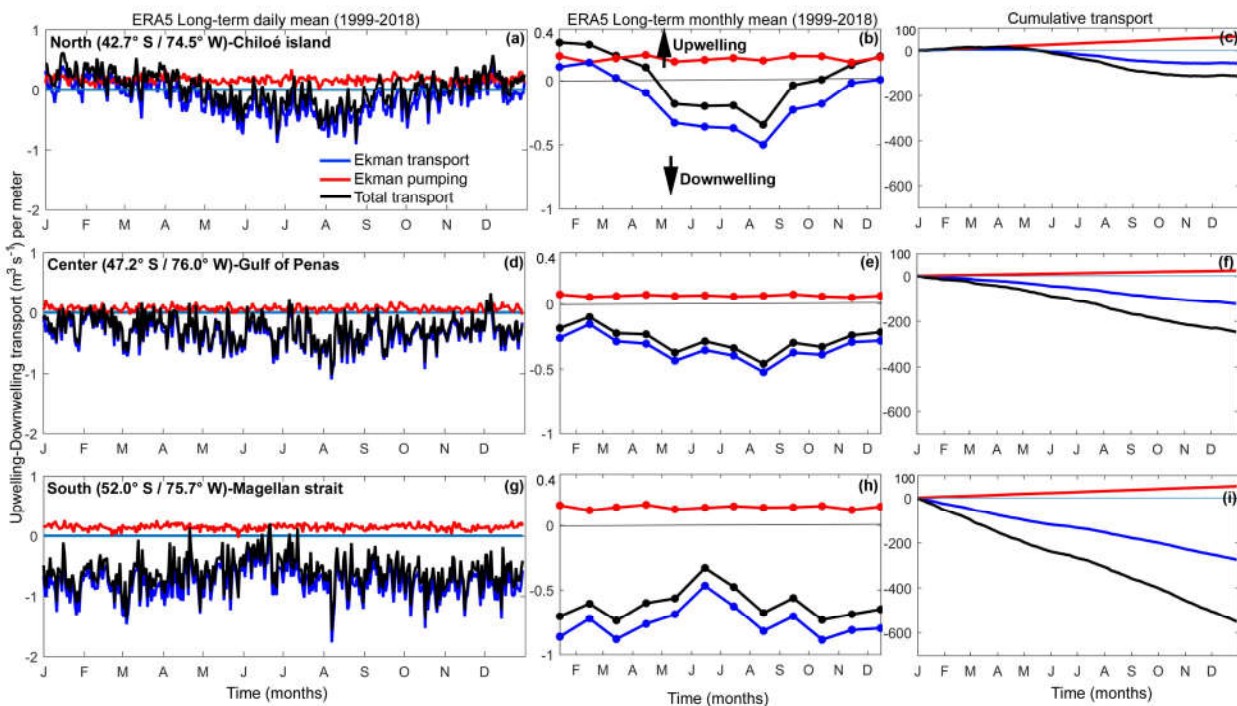

Figure 10. Quantification of the cross-shore transport using ERA5 reanalysis from the north, center, and south time series (see Fig. 1 for the position) from 1999–2018 (**a**, **d**, and **g**) representing the long-term daily mean, (**b**, **e**, and **h**) the long-term monthly mean, and (**c**, **f**, and **i**) cumulative ET, EP, and TUT. The TUT is the sum of the ET and EP. The positive/negative values of transport indicate upwelling/downwelling conditions.

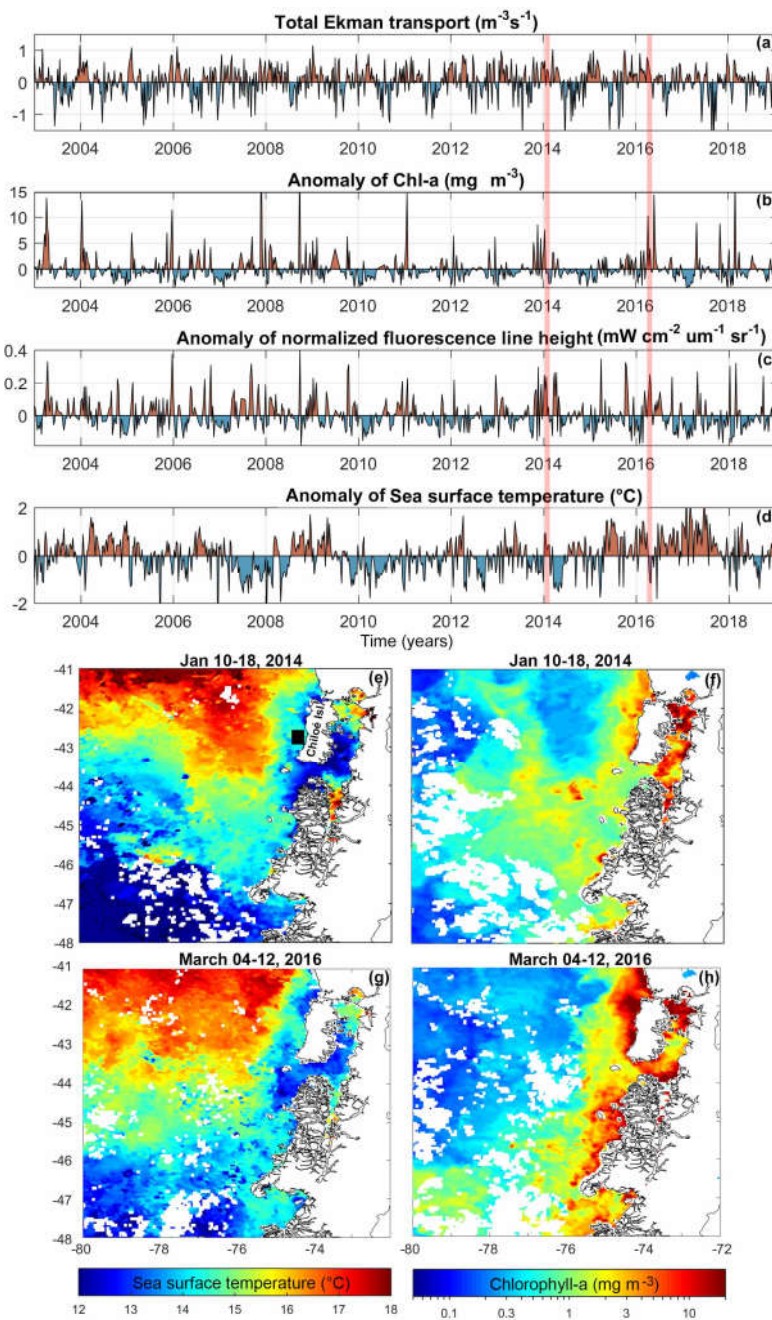

Figure 11. Time series of (**a**) the TUT from ERA5, (**b**) the Chl-a anomalies, (**c**) the FLH anomalies and (**d**) the SST anomalies from the MODIS-Aqua satellite data. (**e-h**) Examples showing the ocean response to ET and EP along the northern coast of Patagonia.  SST (**e** and **g**) and Chl-a (**f** and **h**) from MODIS-Aqua. Time series of TUT (**a**) was obtained from point north of Chiloé Island (see Fig. 10 **a**) and time series from (**b**) Chl-a, (**c**) the FLH, and (**d**) the SST anomalies were extracted from the point closer to the position of TUT time series (solid black square in Fig. 11**e**).

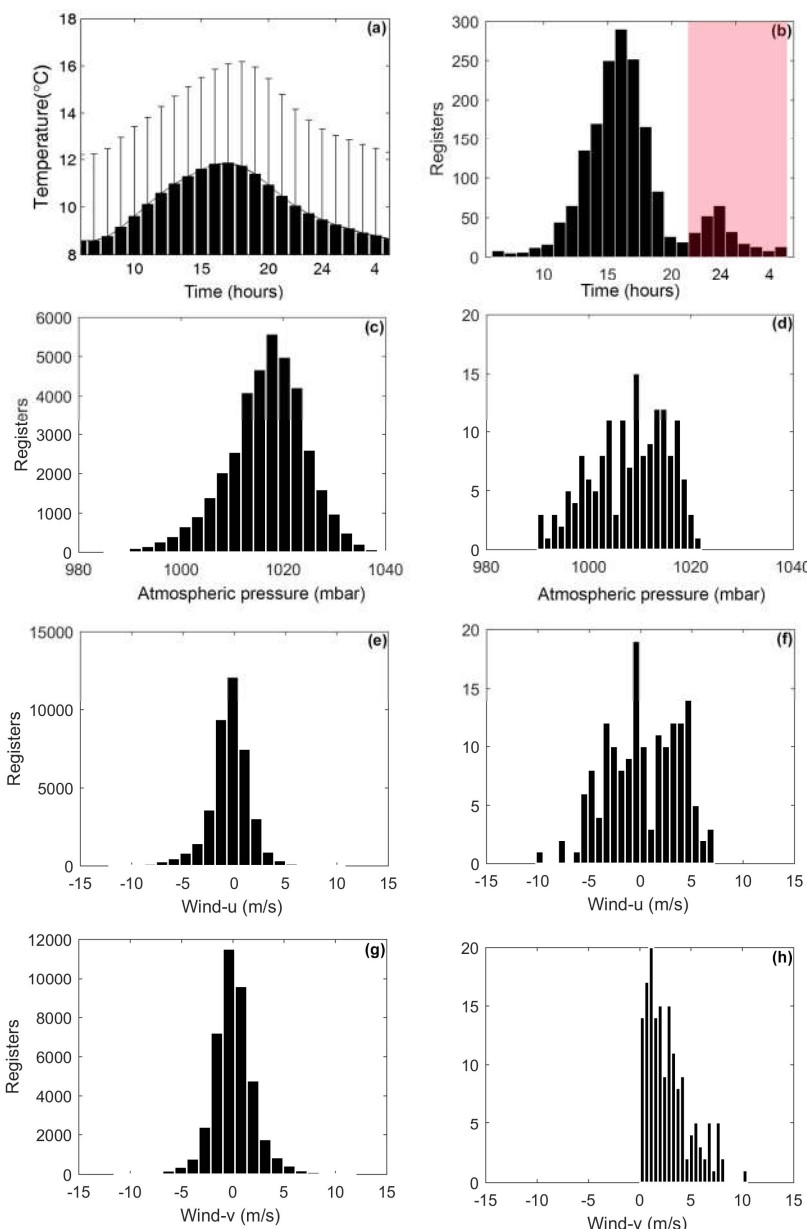

Figure 12. (**a**) SAT long-term hourly means with (**b**) histogram of the maximum SAT. The red shaded area in (**b**) shows the time of the second air temperature maxima. The error bars in (**a**) represent the standard deviations of SAT. (**c**, **e**, and **g**) Complete data set of atmospheric pressure and zonal (wind-u) and meridional wind (wind-v) components. (**d**, **f**, and **h**) Atmospheric pressure and zonal and meridional wind values related to the second SAT maxima. Data were obtained from the Puyuhuapi Fjord oceanographic buoy and meteorological station in the period 2011−2017.

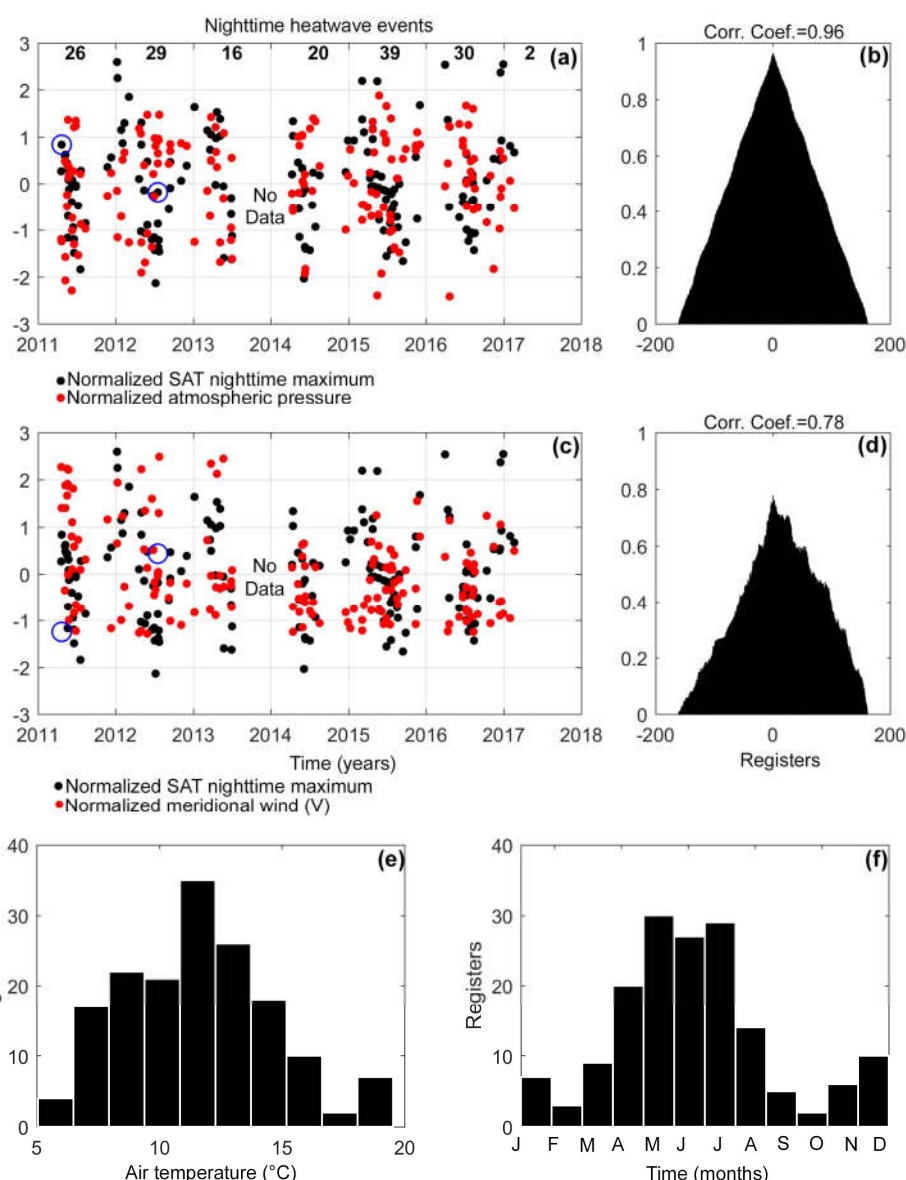

Figure 13. Time series of the nighttime heatwave events. (**a**) Normalized time series of SAT nighttime maximum (black dots) and atmospheric pressure (red dots). (**b**) Cross-correlation coefficient between variables from (**a**). (**c**) Normalized time series of SAT nighttime maximum (black dots) and meridional wind component (red dots). (**d**) Cross-correlation coefficient between variables from (**c**). (**e**) Histogram and (**f**) long-term monthly mean from time

series of SAT nighttime maximum in the period 2011 to 2017. Data were obtained from the Puyuhuapi Fjord oceanographic buoy (2011–2013) and meteorological station (2014–2017). From July 2013 to April 2014, no data were collected. The blue circle in (**a** and **c**) denotes the position of the nighttime heatwave events described in Figs. 14 and 15.

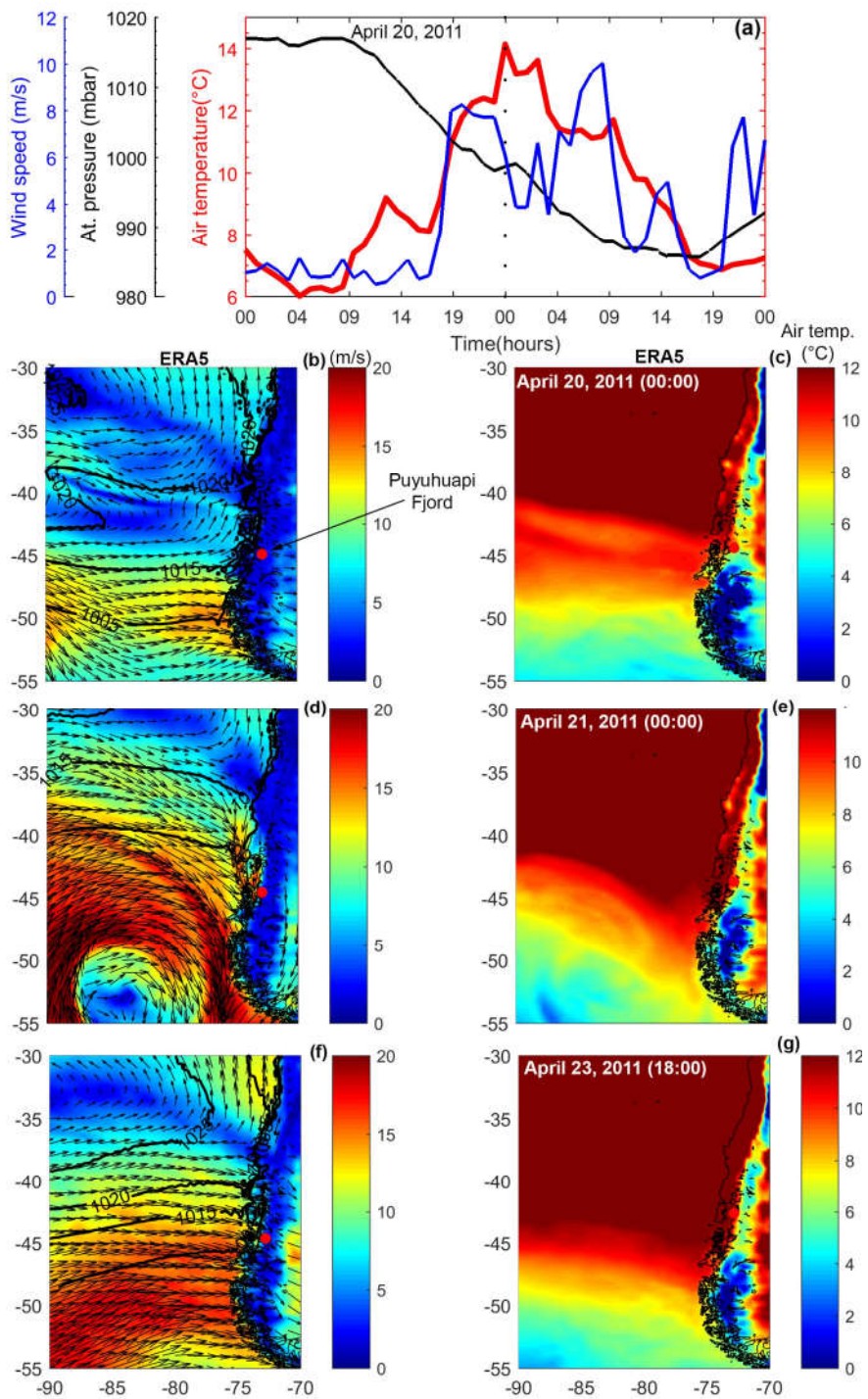

Figure 14. Hourly air temperature, atmospheric pressure, and wind speed data from the Puyuhuapi Fjord oceanographic buoy (**a**) and surface winds, atmospheric pressure and surface air temperature from the ERA5 reanalysis climate product (**b–g**), during April 2011. The surface-wind vectors (**b**, **d**, and **f**) were plotted with a spatial resolution of 1°×1°.

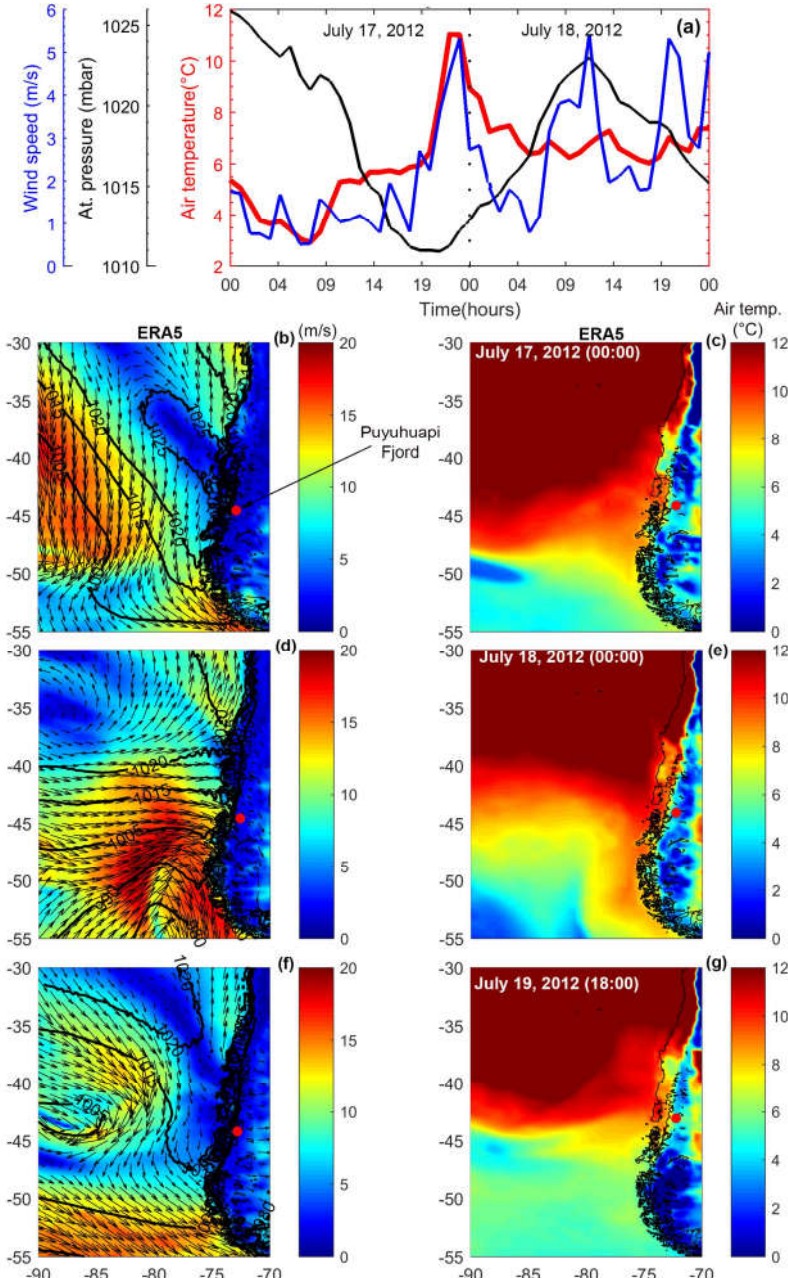

Figure 15. Hourly air temperature, atmospheric pressure, and wind speed data from the Puyuhuapi Fjord oceanographic buoy (**a**) and surface winds, atmospheric pressure and surface air temperature from the ERA5 reanalysis climate product (**b**–**g**), during July 2012. The surface-wind vectors (**b**, **d**, and **f**) were plotted with a spatial resolution of 1°×1°.

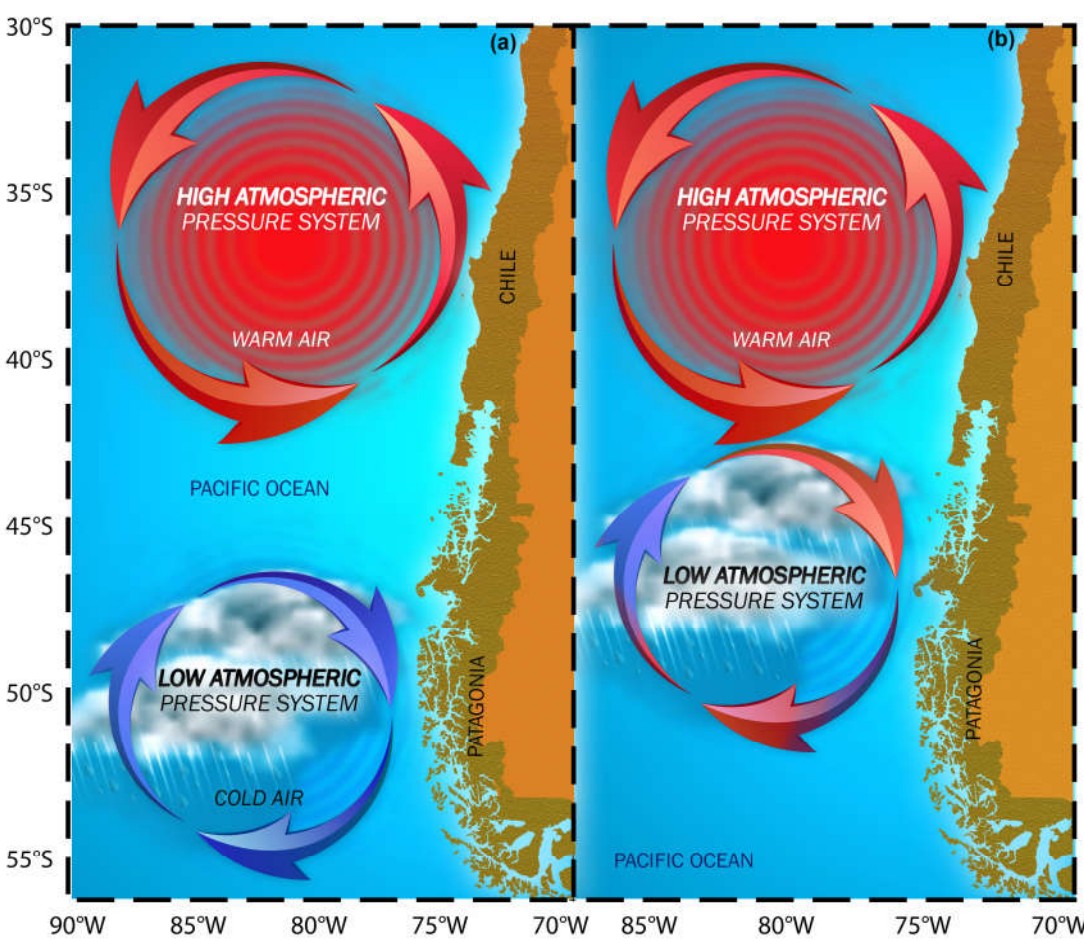

Figure 16. A conceptual model of the "Night-time heatwave event" in the Eastern Austral Pacific Ocean. (**a**) The initial condition, where a low atmospheric pressure system with cold air and a high atmospheric pressure system with warm air are regionally present, although separate; (**b**) the low atmospheric pressure system moves northward and encounters the high atmospheric pressure system, transporting warm air to Patagonia.

830