# Peer review of "Synoptic scale variability of surface winds and ocean response to atmospheric forcing in the eastern Austral Pacific Ocean"

_Ocean Science, 2018_

## Referee Comment (RC1) · Anonymous Referee #1 · 9 Jan 2019

The presented study investigates the variability of southern hemisphere surface winds between 40°S and 56°S. As data basis two scatterometer datasets, Modis chlorophyll measurements, the ERA-Interim reanalysis dataset and observations from meteorological stations and buoys are used. A principal component analysis is applied on the scatterometer data to investigate the first three patterns. Up- and downwelling as well as nighttime heat wave events are investigated. The article is in principle well written. Jumping between the figures (e.g. line 214 fig. 4e than line 215 fig. 2c,f, line 217 fig. 3g,p) in the text makes it harder to follow the argumentation.

The topic of the study is interesting, and I suggest to publish the article after the

following issues are addressed:

- What are the investigated "expected changes" which are mentioned in the title?

- Two different scatterometer datasets (QuickSCAT and ASCAT) are used, which have an overlap of only about two years. The raw satellite data are not gridded. Does the processing of the data can influence the results? Could you give some more details about the data?

- For QuickSCAT, the institution is mentioned from which the data were retrieved (could you change the link to the webpage/ftp where the data are available instead of the institute's main page). It is not mentioned from where you get ASCAT. Are these data treated in the same way as the QuickSCAT data? If not, is there a potential impact on the results?

- In figure 6, a relatively strong difference in the Ekman pumping between both datasets can be seen. Is this because of the different periods of both data sets or are there differences in the observations? In line 169, it is mentioned that for the overlapping period, $R^2$ 0.7. Why wasn't the EOF analysis done also with reanalysis data. This could help to identify the origin of the differences. ERA-Interim assimilates QuickSCAT. There is also a newer reanalysis called ERA5 available, which assimilates also ASCAT. With a resolution of about 0.28°, its resolution is close to the one from the gridded ASCAT data you are using.

- In figure 1, the long term mean for QuickSCAT is higher than for ASCAT. Is this because of the different periods, or has one instrument potentially a bias larger than the other? This could be checked by looking into a homogeneous data set like a reanalysis.

- In line 158 it is mentioned that long term means and linear trends were removed. How? Was it done for each scatterometer data set individually?

- In line 161 it is explained that wavelet spectra were calculated on the entire sampling period. For each data set individually? How are the different resolutions taken into account? In the same way, wouldn't it make sense to repeat the same investigation with a reanalysis?

- You found different cycle lengths for 1999-2008 and 2008-2015. This corresponds more or less to the periods covered by the two scatterometer products. Does the same investigation on reanalysis data would give you comparable results? Or in other words, are the differences related to the two different satellite products and potentially different treatment of the data?

- What are the criteria to identify nighttime heat wave events? The temperature range of the events is specified. Is the definition for example based on the difference between night- and daytime, a heating rate, an exceedance of a threshold or is it only the existence of a second temperature peak at night time?

- The red dashed line in figure 3 (b,e,h,k,n,q) is the 95

- What do the flashes in figure 3 (c,f,l,o) mean ? This is not mentioned in the legend.

- Figure 4: Was ERA-Interim used for the EOF analysis? If this is the case, why are the results not compared to the ones from the observations? Both scatterometer data sets include the date which is shown in the figure.

- Figure 9: It is not explained what the error bars mean. Why is the lower bound not shown?

---

## Short Comment (SC1) · 15 Jan 2019

We thank the reviewer for pointing out all comments, especially the reference to use the ERA-5 product to carried out EOF analysis and the other analysis in the manuscript, e.g., Ekman transport and Ekman pumping. We have started the work with this dataset, but download the data and run all programs to build new figures will take at least one month to the data processing. When we finish with this process, we will submit the new version of the manuscript, including the RC1 comments.

---

## Referee Comment (RC2) · Anonymous Referee #2 · 30 Jan 2019

This paper presents an analysis of wind variability in the eastern Austral Pacific Ocean, and more precisely over the southernmost part of America. Also, the authors look for relationships between wind patterns and the ocean response, as well as the potential impact on nighttime heat waves. I think the goals are interesting and can shed light into the mechanisms behind coastal ocean variability in that region, but I'm concerned about the robustness of their conclusions as the methodology presents some flaws. Probably the intuitions of the authors are right, but a more careful analysis should be performed to support their conclusions.

First issue is about the analysis in two different periods. I understand this is done because of the time coverage of each satellite product, but by doing this it is not clear if the differences reported between periods are due to the period or the product. I think that more efforts should be put in the comparison between products and after calibration use them as a single product and perform the analysis for the whole period. The relationship between the wind structures and the ocean response (SST and Chla) is the most important part of the paper, in my opinion. Therefore should be presented in a more robust way. Using only snapshots is not enough to prove anything. Either you show time series (e.g. SST/Chla evolution against EP/ET or TUT), or use composites (i.e. average the SST for the periods in which HAP/LAP situations are dominant). Something similar happens with the results concerning the nighttime heat waves. Using two hand-picked cases to demonstrate the influence of LAP systems in the nighttime heat waves is not robust. Some statistics as composite images associated with nighttime heat wave periods, or time series analysis would be much better. The separation between Ekman pumping and Ekman transport is interesting. This should probably be discussed in more depth in the discussion section, as well as the implications the different components may have on the ocean evolution.

Detailed comments The title doesn't seem adequate. The ocean-atmosphere coupling is not taking into account (only atmosphere forcing on ocean). Also no expected changes are analysed.

I miss an introductory figure with the map of the zone of interest with the major wind patterns. For the introduction it would be useful to clearly state if HAP and LAP are symmetric atmospheric situations. L90-94: I think this does not fit as a final sentence for the introduction and should be moved elsewhere. L105. Has ERA-Interim or the satellite data been validated in this region? This is important as the quality of those products is not the same everywhere. If you have wind data from local stations it would be worth comparing them with it to assess the quality at different time scales. L110. Show the location of the stations in an introductory figure. L118. It is not clear if the data is 15-mins or hourly. Section 2.4. It would be useful to briefly describe what is

Ekman transport and Ekman pumping, what physical process involves, for the non-oceanographers. L142-145. Why can't you compute the curl? Can't you use the wind over land for that? Or alternatively, a 0 wind? Also, I'm not sure your choice of extrapolating the wind curl to the near-coast points is better. Although there is probably not a best option you should discuss the implications of that extrapolation in your results, as you may be overestimating the Ekman pumping near the coast. L148. It would be better to show the sections you are using in an introductory figure. In Fig 5 is not clear at all. L156-157 "This method .... " This sentence is not needed. L158. "..the three LEADING modes" L163. The hourly, daily and monthly means are exactly the same. If you refer to computing the means for each hour, then you should explain it better. L163-171. I think this paragraph is repetitive with ideas presented before and can be rewritten. L168. Time correlation is not a statistical moment. Also, you should compare the differences in magnitude (e.g. STD) and the RMSE. Also time correlation should be computed for the different data sampling you analyse here (e.g. hourly, daily or monthly). L174. Why the period 1999-2015? It doesn't match with the period covered by the products. L193-196. Beware, EOF analysis works on anomalies, so they reflect weakenings or strenghtenings of the mean field, and may not mean a change in the direction of the total wind field. Please, reconsider your statement. Figure 3. What do the arrows mean?. Figure 4 is strange. Here there are more than one EOF acting. If not, EOF+ and EOF- should be exactly the opposite. I think this figure,as it is is more confusing. L257-259. It is not clear that ET/EP are strong in those examples. Probably showing time series of ET/EP would be more illustrative than single snapshots. Also, about the maps, they are confusing, too much information there. Probably a simpler figure with the wind field and the TUT in colors would be clearer. L268-270. This sentence doesn't seem very relevant in this context. Again it would be better to show the location of the station in an introductory figure. L273-274. I don't think the histograms are enough to prove the solar radiation forces the diurnal cycle. Although is probably the case , correlations or explained variance diagnostics would be better suited for that. Figure 9. It is not clear what the bottom panels represent. Figure 10. The caption is

not clear. L345-354. This is not supported by any result shown in the paper. Either the authors show new figures or remove this.

---

## Author Comment (AC1) · 6 Feb 2019

Dear Referee

We appreciate very much all your comments to the manuscript. We have started to work in order to better the quality of the manuscript.

Best, Iván
* * *

---

## Author Comment (AC2) · 29 Mar 2019

The presented study investigates the variability of southern hemisphere surface winds between 40_S and 56_S. As data basis two scatterometer datasets, Modis chlorophyll measurements, the ERA-Interim reanalysis dataset and observations from meteoro-

logical stations and buoys are used. A principal component analysis is applied on the scatterometer data to investigate the first three patterns. Up- and downwelling as well as nighttime heat wave events are investigated. The article is in principle well written. Jumping between the figures (e.g. line 214 fig. 4e than line 215 fig. 2c,f, line 217 fig.3g,p) in the text makes it harder to follow the argumentation. The topic of the study is interesting, and I suggest to publish the article after the following issues are addressed:

General comment to RC1: We appreciate the recommendation of the reviewer to add the ERA5 climate data set to the manuscript and also carried out the statistical analysis, e.g., EOF and wavelet. New figures were added to the manuscript contributing to an increase in the quality and strength of the results and discussion presented in the new manuscript version.

 c What are the investigated "expected changes" which are mentioned in the title?

We modified the title to, "Synoptic scale variability of surface winds and ocean response to atmospheric forcing in the eastern Austral Pacific Ocean".

 c Two different scatterometer datasets (QuickSCAT and ASCAT) are used, which have an overlap of only about two years. The raw satellite data are not gridded. Does the processing of the data can influence the results? Could you give some more details about the data?

The two scatterometers datasets overlap in the information on the surface winds from 2007 to 2009. We extracted three time series for the zonal and meridional wind components in areas close to coastal zone (e.g., 42.2° S, 47.2° S and 52.2° S, see the new figure 1 to the geographical position). The linear regression between both scatterometers was high with an r2 range from 0.65 to 0.73. The raw data for each dataset are now presented in the new figure 3, where the raw data from the ERA5 reanalysis is also presented.

• For QuickSCAT, the institution is mentioned from which the data were retrieved (could you change the link to the webpage/ftp where the data are available instead of the institute's main page). It is not mentioned from where you get ASCAT. Are these data treated in the same way as the QuickSCAT data? If not, is there a potential impact on the results?

We added the ftp website for both satellite surface wind products: QuikSCAT (ftp.ifremer.fr/ifremer/cersat/products/gridded/mwf-quikscat/data/) and ASCAT (ftp.ifremer/cersat/products/gridded/MWF/L3/ASCAT/). The ASCAT database was treated similarly ro QuikSCAT. The only difference was in the spatial resolution.

• In figure 6, a relatively strong difference in the Ekman pumping between both datasets can be seen. Is this because of the different periods of both data sets or are there differences in the observations?

We agree with this comment. The new calculation of Ekman pumping using the ERA5 data set (new figure) exhibited a similar behavior to the results of QuikSCAT. The Ekman pumping values are a good representationof the features in the three products (QuikSCAT, ASCAT, and ERA5 reanalysis), e.g., the area between 50°-54° S / 75°-80° W and the coastal zone between latitudes 40°-44° S. We added these new results and discussion to the manuscript.

• In line 169, it is mentioned that for the overlapping period, R2 0.7. Why wasn't the EOF analysis done also with reanalysis data. This could help to identify the origin of the differences. ERAInterim assimilates QuickSCAT. There is also a newer reanalysis called ERA5 available, which assimilates also ASCAT. With a resolution of about 0.28_, its resolution is close to the one from the gridded ASCAT data you are using.

As was recommended, we added the ERA5 dataset to the manuscript. This new data was used in the EOF calculation and other analyses. New figures were generated, and in general, the results obtained with ERA5 agree with the results obtained with QuikSCAT and ASCAT.

⇢ In figure 1, the long term mean for QuickSCAT is higher than for ASCAT. Is this because of the different periods, or has one instrument potentially a bias larger than the other? This could be checked by looking into a homogeneous data set like a reanalysis.

We calculated the long term mean using the ERA5 reanalysis data set over the period 1999-2015. The new result is presented in figure 2 and is similarto the ASCAT long term mean. We added a new description of figure 2 in the text.

⇢ In line 158 it is mentioned that long term means and linear trends were removed. How? Was it done for each scatterometer data set individually?

Yes, we eliminated the linear trends individually for QuikSCAT, ASCAT and ERA5.

⇢ In line 161 it is explained that wavelet spectra were calculated on the entire sampling period. For each data set individually? How are the different resolutions taken into account? In the same way, wouldn't it make sense to repeat the same investigation with a reanalysis?

The wavelet spectra analysis was applied individually for each scatterometers, e.g., for QuikSCAT over the period 1999-2009 and ASCAT over the period 2007-2015. We repeated the same analysis for the ERA5 reanalysis data. The new data set and results were added to the manuscript. The ERA5 covered the entire sampling period 1999-2015 and showed similar results to QuikSCAT and ASCAT. We clarified the sentence in line 161.

⇢ You found different cycle lengths for 1999-2008 and 2008-2015. This corresponds more or less to the periods covered by the two scatterometer products. Does the same investigation on reanalysis data would give you comparable results? Or in other words, are the differences related to the two different satellite products and potentially different treatment of the data?

The wavelet analysis carried out with the ERA5 reanalysis data set confirmed the different cycle lengths observed with the QuikSCAT and ASCAT data set.

• What are the criteria to identify nighttime heat wave events? The temperature range of the events is specified. Is the definition for example based on the difference between night- and daytime, a heating rate, an exceedance of a threshold or is it only the existence of a second temperature peak at night time?

We explored different statistic tools to automatically identified the "nighttime heat wave events," based on the spectral analysis, the dominance periods of 12 and 24 hours were extracted from the time series but the residual time series did not clearly show the events. Therefore, the best tool was selecting the daytime that we observed the occurrence of the process, and with a detailed manual validation, the nighttime heat wave events were quantified. In the future, the additional effort will be put into finding an efficient tool. We believe that the reports and publication of this new event will attract the attention of the scientific community and new tools will facilitate development.

• The red dashed line in figure 3 (b,e,h,k,n,q) is the 95.

Yes, The red dashed line in figure 3 (b,e,h,k,n,q) is the 95

• What do the flashes in figure 3 (c,f,l,o) mean ? This is not mentioned in the legend.

The arrows in figure 3 (c, f, l and o) indicated the normalized eigenvector patterns presented in figure 2 (a, b, d and e). We decided to eliminated the arrows from the figure.

• Figure 4: Was ERA-Interim used for the EOF analysis? If this is the case, why are the results not compared to the ones from the observations? Both scatterometers data sets include the date which is shown in the figure.

As we mentioned before, a new data set for the ERA5 reanalysis was added to the manuscript covering the entire sampling period (1999-2015). The EOF result was added to a new figure and also to the text, showing similar variance and eigenvector behavior as QuikSCAT and ASCAT.

• Figure 9: It is not explained what the error bars mean. Why is the lower bound not
shown?

In figure 9, the error bars denote the standard deviation. As the lower and upper bounds have the same values, we decided to plot the upper bound only. We clarified the figure 9 caption and new text was inserted in the new version of the manuscript.

———————————————

---

## Author Comment (AC3) · 29 Mar 2019

This paper presents an analysis of wind variability in the eastern Austral Pacific Ocean, and more precisely over the southernmost part of America. Also, the authors look for

relationships between wind patterns and the ocean response, as well as the potential impact on nighttime heat waves. I think the goals are interesting and can shed light into the mechanisms behind coastal ocean variability in that region, but I'm concerned about the robustness of their conclusions as the methodology presents some flaws. Probably the intuitions of the authors are right, but a more careful analysis should be performed to support their conclusions.

General comment to RC2: We appreciate all recommendations of the reviewer, especially the addition the time series of different variables and processes, such as those related to the Ekman upwelling quantification and the ocean response. The new total Ekman upwelling quantification demonstrated the dominance of Ekman pumping instead of the Ekman transport as was proposed before and recently by other authors. Additionally, supplement material has been added which presents different figures associated with the validation processes between wind satellite and reanalysis products with in-situ data (buoys and navy lighthouse). These analyses demonstrated the high correlation and low root mean square error and standard deviation between in-situ data and the ERA5 reanalysis climate data set.

First issue is about the analysis in two different periods. I understand this is done because of the time coverage of each satellite product, but by doing this it is not clear if the differences reported between periods are due to the period or the product. I think that more efforts should be put in the comparison between products and after calibration use them as a single product and perform the analysis for the whole period.

-In order to validate the results obtained with the two scatterometers, the ERA5 reanalysis data set was incorporated into the manuscript. The ERA5 covered the complete and continuous sampling period of satellites (1999-2015). The data analysis of ERA5 confirmed and validated the results showed by the scatterometers. New figures and text were added to the manuscript.

The relationship between the wind structures and the ocean response (SST and Chla)

is the most important part of the paper, in my opinion. Therefore should be presented in a more robust way. Using only snapshots is not enough to prove anything. Either you show time series (e.g. SST/Chla evolution against EP/ET or TUT), or use composites (i.e. average the SST for the periods in which HAP/LAP situations are dominant).

-As was recommended by RC2, we extracted a time series of the Total Ekman Transport, same as TUT, along with the west coast of Chiloé island, where favorable upwelling conditions were observed. The ERA5 data set was used for this calculation. Time series of Chl-a, normalized fluorescence line height and SST from MODIS AQUA were used in a temporal and spatial resolution of 8 days and 4 km. A new figure is presented in the manuscript to show the precise relationship between wind structures and the ocean response during the period 2002-2018.

Something similar happens with the results concerning the nighttime heat waves. Using two hand-picked cases to demonstrate the influence of LAP systems in the nighttime heat waves is not robust. Some statistics as composite images associated with nighttime heat wave periods or time series analysis would be much better.

-We added a new subplot figure (Fig. 13) that shows the time series of the nighttime heat wave events. Also, a correlation process was applied between air temperature and the atmospheric pressure of each of the events revealing a high correlation coefficient.

The separation between Ekman pumping and Ekman transport is interesting. This should probably be discussed in more depth in the discussion section, as well as the implications the different components may have on the ocean evolution.

-We added more information and a discussion of the relevance of Ekman upwelling in the ocean response. New references were incorporated, and the new analysis and quantification of total upwelling demonstrated the dominance of Ekman pumping instead of Ekman transport along the western coastline of Chiloé island.

Detailed comments

The title doesn't seem adequate. The ocean-atmosphere coupling is not taking into account (the only atmosphere forcing on the ocean). Also, no expected changes are analyzed.

-We modified the title, to "Synoptic scale variability of surface winds and ocean response to atmospheric forcing in the eastern Austral Pacific Ocean".

I miss an introductory figure with the map of the zone of interest with the major wind patterns. -We added a new figure 1 to show the geographical position of stations and analysis.

For the introduction, it would be useful to clearly state if HAP and LAP are symmetric atmospheric situations.

-We incorporated new information in the Introduction.

L90-94: I think this does not fit as a final sentence for the introduction and should be moved elsewhere. -We eliminated the final sentence of the introduction section.

L105. Has ERA-Interim or the satellite data been validated in this region? This is important as the quality of those products is not the same everywhere. If you have wind data from local stations it would be worth comparing them with it to assess the quality at different time scales. -We added supplemental material that incorporates different figures of the validation process carried out between satellites and reanalysis surface wind products with in-situ local stations, such as, buoy and navy lighthouse. A Taylor diagram was applied showing satisfactory results.

L110. Show the location of the stations in an introductory figure.

-As we mentioned before, a new figure 1 was added to the manuscript.

L118. It is not clear if the data is 15-mins or hourly. - We have clarified the information in the text. The raw atmospheric data from the buoy (3 minutes) and the meteorological station (15 minutes) were hourly averages.

Section 2.4. It would be useful to briefly describe what is Ekman transport and Ekman pumping, what physical process involves, for the non-oceanographers.

-We added a new paragraph in section 2.4 that describes the importance of Ekman transport and pumping as physical processes to the biology and in general for non-oceanographers.

L142-145. Why can't you compute the curl? Can't you use the wind over land for that? Or alternatively, a 0 wind? Also, I'm not sure your choice of extrapolating the wind curl to the near-coast points is better. Although there is probably not the best option you should discuss the implications of that extrapolation in your results, as you may be overestimating the Ekman pumping near the coast.

We computed the wind stress curl as shown in equation 5. In the computation of the wind stress curl, only the data available over the sea was used. During the wind stress curl calculation, the closest grid point to the coast is lost. The extrapolation process thus only incorporated this point in the data. We compared the wind stress curl calculation with and without the extrapolation processes and the results did not change. The positive Ekman pumping velocities registered in the coastal zone, especially in the northern domain, extended for more than five grid point into the ocean. We decided to continue with this methodology, but the results obtained in the interior fjords and channels were deleted in the new figure.

L148. It would be better to show the sections you are using in an introductory figure. In Fig 5 is not clear at all. -As we mentioned before, a new figure 1 has been added to the manuscript.

L156-157 "This method .... " This sentence is not needed. -We eliminated the sentence.

L158. "..the three LEADING modes" -We added "leading" to the sentence.

L163. The hourly, daily and monthly means are exactly the same. If you refer to

computing the means for each hour, then you should explain it better.

-We eliminated this sentence from the text.

L163-171. I think this paragraph is repetitive with ideas presented before and can be rewritten.

-We eliminated this sentence from the text.

L168. Time correlation is not a statistical moment. Also, you should compare the differences in magnitude (e.g. STD) and the RMSE. Also, time correlation should be computed for the different data sampling you analyze here (e.g. hourly, daily or monthly). -Correlation process and Taylor diagram were applied in all cases where necessary. New results and discussion was added to the manuscript.

L174. Why the period 1999-2015? It doesn't match with the period covered by the products.

-We agree with the comments, but the manuscript was written some years ago, with the data set available at this time. In this version, we have incorporated the ERA5 reanalysis data set to December 2018, with which we demonstrated the similarities to QuikSCAT and ASCAT. The ocean response to the surface wind using the derived parameters, e.g., Ekman pumping and transport were presented for the period reported by ERA5 reanalysis (1999-2018).

L193-196. Beware, EOF analysis works on anomalies, so they reflect weakenings or strengthenings of the mean field, and may not mean a change in the direction of the total wind field. Please, reconsider your statement.

We have clarified this sentence "The spatial structure for the first three modes from the QuikSCAT and ASCAT databases were similar (Fig. 2). In the case of the spatial structure of mode 1 (Fig. 2a and 2d), southerly and southwesterly winds dominated the study area, when the time-dependent coefficient was positive (Fig. 3a and 3j, PC-1). When PC-1 (principal component) was negative, the spatial structure of mode 1

changed the direction, and northerly and northeasterly winds occurred".

In this manuscript, we are using the same EOF method (real-vector EOF) proposed in Kaihatu et al., (1998). In the description of the methodology, the authors wrote: "The time-dependent coefficients show the magnitudes and directions of the vectors; negative coefficients denote a 180° shift in the direction relative to that shown on the spatial map".

Figure 3. What do the arrows mean?.

The arrows in figure 3 (c, f, l, and o) indicated the normalized eigenvector patterns presented in figure 2 (a, b, d and e). We have decided to eliminate the arrows from the figure.

Figure 4 is strange. Here there are more than one EOF acting. If not, EOF+ and EOF- should be exactly the opposite. I think this figure, as it is more confusing.

-We have clarified the information in figure 4, but we believe that figure 4 is important to the manuscript because it is the only figure that shows different examples of the influence of HAB and LAP systems in the study region.

L257-259. It is not clear that ET/EP is strong in those examples. Probably showing time series of ET/EP would be more illustrative than single snapshots. Also, about the maps, they are confusing, too much information there. Probably a simpler figure with the wind field and the TUT in colors would be clearer.

-We eliminated ET/EP examples and added the time series of the Total Ekman Transport together with the ocean response variables, e.g., Chl-a, SST, etc. A new figure is present in the manuscript.

L268-270. This sentence doesn't seem very relevant in this context. Again it would be better to show the location of the station in an introductory figure. -We eliminated the first sentence from the text and added the location of the stations in figure 1.

L273-274. I don't think the histograms are enough to prove the solar radiation forces the diurnal cycle. Although is probably the case, correlations or explained variance diagnostics would be better suited for that.

-We carried out a new statistical analysis to prove the relationship between variables. New subplot figures were added.

Figure 9. It is not clear what the bottom panels represent.

-The bottom panels in figure 9 (e and f) represent the histogram of the maximum surface air temperature as we mentioned in the caption of figure 9.

Figure 10. The caption is not clear.

-The information presented in figure 10 shows the results from the nighttime heat wave events. We improved the caption of figure 10 and also included more details inside the figure.

L345-354. This is not supported by any result shown in the paper. Either the authors show new figures or remove this. -We removed lines 345-354 from the text and also the reference includes in this paragraph (Alvarinho et al., 2006).

Please also note the supplement to this comment:
https://www.ocean-sci-discuss.net/os-2018-119/os-2018-119-AC3-supplement.pdf

---

## Author Response (AR2)

p. 9 l. 288 covering1999

We eliminated the error from the text.

p. 14 l. 460 irregular orographic from. This have to be corrected before publication. It is not mentioned from where the authors got the Quikscat and ASCAT data. It should also be mentioned in the manuscript like in the answer to reviewer 1 that both scatterometer datasets were treated in a similar way.

In this paragraph when we reference to the irregular orographic, means that coastal line is irregular (not a continue line like between 42°-43° S) due to the presence of many islands separate by channels. In any case we reference to the QuikSCAT and ASCAT scatterometers data sets.

We clarify the sentence as fallow "We have hypothesized that the irregular orographic structure of the coastline from 44°S–56° S, where the coast is comprised of many islands and channels, could reduce the possibility of the oceanic water sinking at the coastline passing into the interior of the Patagonian fjords and carrying the eggs and larvae of many species and nutrients and enhancing biological production".

**Interactive comment on**
**"Synoptic scale variability of surface winds and ocean response to atmospheric forcing in the eastern Austral Pacific Ocean"**

**Anonymous Referee #2**

I appreciate the efforts the authors have made to answer my comments, in particular the addition of ERA5 data and the time series of Chl-a and SST. The new analysis of ERA5 allows to clarify that the different behavior between QUICKScat and ASCAT is not due to instrumental issues but to variability (Figure 8). Nevertheless there are a couple of important issues that should be addressed. In general I think the results do not support robust conclusions. I understand that it could be difficult and that maybe other statistical approaches should be followed. Therefore, in general I would suggest to moderate the conclusions. For instance, the results indicate that LAP/HAP system may affect the nightime heatwaves, but it is by no means demonstrated.

-We moderate conclusions.

In the analysis of the EOFs is stated (L 240) that "The southerly and northerly winds were associated with the passage of intense HAP (Fig. 7a) and LAP (Fig. 7b) systems, throughout the study region." and in (L 396) " LAP and HAP systems dominated mode 1 of the EOF, contributing 30 % of the total variance (Fig. 3-6). In this mode, southerlies related to the passage of HAP systems, and northerlies produced by LAP systems (Fig. 7)". But this has not been demonstrated at all. The EOF patterns do not show anything similar to a HAP/LAP structure. This should be rewritten throughout the paper.

- A new EOF analysis was implemented in order to demonstrate the influence of HAP/LAP systems in the EOF patterns. In this calculation, the same ERA5 reanalysis data set was used, but the study area was enlarged. The EOF results were incorporated in the supplementary material document as figures S6, S7, and S8. The new EOF patterns showed similarities with previous EOF calculations presented in Fig. 3 of the last version of manuscript submitted and confirmed that the southerly and northerly winds were associated with the passage of HAP and LAP systems for the region. Therefore, we sustain our conclusions.

A new sentence was added to the manuscript:

"To capture the influence of the LAP and HAP systems in the EOF patterns, the ERA5 data set was used to carry out a further EOF analysis, but this time the study region was expanded to the west (120° W) and the north (30° N). This EOF analysis confirmed our previous conclusions (See Supplementary Material, Fig. S6–S7)."

The comparison of time series of Chl-a and SST with respect to Ekman Transport (Figure 11) shows relatively low correlation (section 3.2), with values of around 0.3. Therefore, it can't be concluded that EP/ET are the responsible of Chl-a and SST variations. This should be acknowledged in the abstract and discussion.

-In the abstract and conclusion section we comment that EP/ET contributed/favorite with the reduced/decreased of Chl-a and SST, but we don't confirmed 100 % that both processes (EP/ET) were responsible by the variations of Chl-a and SST in the coastal zone.

-We also clarify in the discussion section the contribution of EP/ET to the Chl-a and SST variations.

In Section 3.3 the figure 12 is analysed. I repeat my comment, I don't think the figure demonstrates solar radiation drives the diurnal cycle of SAT. The maximum is shifted: maximum solar radiation is at 14h while maximum SAT is at 16h. In any case, the interest of this section is not the diurnal maximum but the night heat waves, so that part could be removed.

-We eliminated from the text and the figure 12, the information from the solar radiation data.

Regarding night heat waves, the mechanism driving them is not clear, as was pointed out in my previous review. Correlation between atmospheric pressure and air temperature (Figure 13), seems too high for what is shown in the figure. It is difficult to compare both time series as it is presented. Maybe plotting both time series in the same plot (i.e. normalized or using different y axis) would help to clearly show that both time series are correlated. As it is shown now it is hard to see that both time series are correlated at the 0.96 level as is stated in the figure title.

-We modified Figure 13. Time series of SAT nighttime maximum, atmospheric pressure and meridional winds were normalized, and the new plots shower the relationship between variables. The correlation coefficient results were added to the figure.

Moreover, the relating SAT and atmospheric pressure is not probably the right way to confirm that winds are the responsible of night heatwaves. Pressure is only related to winds through the gradients, not the absolute value. In the previous review I suggested to use some statistics as composite images associated with nighttime heat wave periods or to show histograms of wind components in the periods when night heatwaves occured. In any case, to show only two hand-picked cases is not robust enough to conclude that LAP systems are the responsible for night heatwaves, as is stated in the abstract.

-We modified Figure 12. In the new figure, histograms analysis of the atmospheric pressure and zonal and meridional winds were incorporated for the moment of occurrence of nighttime heatwaves. We also added other studies cases in the supplementary material (Fig. S9) to confirm the occurrence of nighttime heatwave events in where LAP systems were involved.

Detailed comments:

Abstract. It is stated that between 41-43ºS Ekman Pumping dominates in spring-summer, but from Figure 10 it looks the opposite. Total Transport is clearly driven by Ekman Transport, but in spring-summer it changes its sign with respect to what happens in the other locations/seasons. In any case, Ekman Pumping is similar in the three locations and secondary.

-We changed the bar subplots of Figure 10 (**b**), (**e**) and (**h**) by a line subplots in order to showed better the long-term monthly mean of ET, BE and TUT. We agree with the reviewer that in general TUT was dominated/modulated by the ET, but Ekman pumping also contributed to the TUT during spring-summer in the northern point located offshore Chiloé Island and need to be take into the account in the TUT quantifications.

-We edited the sentence in the abstract as, "In the zonal band between 41°–43° S, the latitude of Chiloe Island, upward Ekman pumping and Ekman transport during spring and summer favored a reduced sea surface temperature and increased chlorophyll-a levels; this is the first time that such Ekman upwelling conditions have been reported so far south in the eastern Pacific Ocean."

The computation of the wind curl is only done over the sea, while in coastal areas this means losing data points. I think the authors could compute it with ERA5 to get something more reliable in the coastal area.

-We calculated again the wind stress and wind stress curl using a new ERA5 mask, which incorporated more data point closer to the coastal line. The results from wind stress showed an abrupt decreased in wind intensity in the coastal areas adding a big error in the wind stress curl computations. As ERA5 reanalysis data set used the input of different satellite scatterometers (e.g., ERS, ASCAT, QuikSCAT) in the process of the data base construction, wind data over the land underestimated wind magnitude and direction (see data over land in figure 7 from manuscript submitted). Therefore, we decided didn't added this results to the new version of the manuscript and then we maintain results presented in the last version.

Figure 11, the figure caption should mention from which point are the time series extracted.

-We added a sentence to the figure caption. "Time series of TUT (**a**) was obtained from point north of Chiloé Island (see Fig. 10 **a**) and time series from (**b**) Chl-a, (**c**) the FLH, and (**d**) the SST anomalies were extracted from the point closer to the position of TUT time series (solid black square in Fig. 11**e**)."

In figure 12 (before figure 9), I don't understand yet what panels g and h represent. The authors haven't done any modification on that, in spite of what they answer in the reply. By elimination I deduce they refer to maximum SAT , but it is not clear what maximum refers to. Is it the maximum measured in the whole observational period? The daily maximum?.

-We modified Figure 12, as we mentioned before.

[revised manuscript text omitted]
 passinginto the interior of the Patagonian fjords,and carryingthe nutrients, eggs,and larvae from of many species andnutrients andinto these areas,toenhance biological production in the southern Patagonian fjords.enhancing biological production.

IIn addition, it was noted that it was not only the ocean that responded to the synoptic synoptic-scale variability of
500 the surface wind,;butthatthe atmospheric conditions were also influenced.This studyregistered approximatelyA total of ~160 events were registered in this manuscript,in which the SAT nighttime maximum ("Nnighttime heat waveheatwave events") occurred in response tothe influence of low atmospheric pressure systems with predominant winds from the northwest and northeast directions predominating, registering a high correlation coefficient between the SAT nighttime maximum with the atmospheric pressure and meridional wind components (Fig. 12 and Fig. 13).
505 Various examples demonstrated the importance of the synoptic synoptic-scale events in modifying climate conditions in the Austral region (Fig. 14,Fig. 15 and Fig. S9), where the LAP systems contribute withto the origin of the nighttime heat waveheatwave events.

A conceptual model was built to explain the source of the nighttime heat waveheatwave events (Fig. 16). In this model, two atmospheric pressure systems participated: a permanent high pressure located in the north (SPSA),
510 which transported warm air from the subtropical region (over the 40° S), and a synoptic LAP system, whichoriginated in the south, with cold air from the Polar zone (Fig. 16a). The LAP originated in the Austral-Pacific Ocean, and the system moved northward, with intense winds rotating clockwise. The northward-moving LAP stopped when it encountered the southern edge of the SPSA,at approximately 40° S (Fig. 16b).At this momentThen, the stronger west and northwest winds from the LAP pulled in the warm air from the SPSA, and advectedits heat
515 southward to Patagonia. These events occur more frequently atnighttime, and theirimpact on the Patagonian climate depends on the intensity of the LAP system winds,and the heat content of the SPSA.

In the contexts of climate change and variability, anyincrease or trend of changechanging trends in these events needs toshouldbe taken into accountconsidered, as mechanisms that could contribute toincreasedglacial 
[revised manuscript text omitted]